# Semi-supervised Graph Anomaly Detection via Robust Homophily Learning

**Guoguo Ai**[1,2*], **Hezhe Qiao** [2*], **Hui Yan** [1†], **Guansong Pang** [2†]

[1]School of Computer Science and Engineering, Nanjing University of Science and Technology
[2]School of Computing and Information Systems, Singapore Management University
`{guoguo, yanhui}@njust.edu.cn,`
`hezheqiao.2022@phdcs.smu.edu.sg, gspang@smu.edu.sg`

## Abstract

Current semi-supervised graph anomaly detection (GAD) methods utilizes a small set of labeled normal nodes to identify abnormal nodes from a large set of unlabeled nodes in a graph. These methods posit that **1)** normal nodes share a similar level of homophily and **2)** the labeled normal nodes can well represent the homophily patterns in the entire normal class. However, this assumption often does not hold well since normal nodes in a graph can exhibit diverse homophily in real-world GAD datasets. In this paper, we propose **RHO**, namely Robust Homophily Learning, to adaptively learn such homophily patterns. RHO consists of two novel modules, adaptive frequency response filters (**AdaFreq**) and graph normality alignment (**GNA**). AdaFreq learns a set of adaptive spectral filters that capture different frequency components of the labeled normal nodes with varying homophily in the channel-wise and cross-channel views of node attributes. GNA is introduced to enforce consistency between the channel-wise and cross-channel homophily representations to robustify the normality learned by the filters in the two views. Experiments on eight real-world GAD datasets show that RHO can effectively learn varying, often under-represented, homophily in the small labeled node set and substantially outperforms state-of-the-art competing methods. Code is available at https://github.com/mala-lab/RHO.

## 1 Introduction

Graph anomaly detection (GAD) has received increasing attention in recent years due to its wide range of applications, such as fraud detection in finance, review spam detection, and abusive user detection [1, 20, 27, 37]. Semi-supervised GAD, which aims to leverage a small set of labeled normal nodes to identify abnormal nodes from a large set of unlabeled nodes in a graph, is among the most realistic problem settings in real-life applications [33, 37, 38]. This is because GAD datasets are typically dominated by normal nodes, which can greatly facilitate the annotation of a few normal nodes, *e.g.*, if we randomly sample a few nodes from a large graph, most nodes would be normal. Current semi-supervised GAD methods are generally built upon two key assumptions: (1) all normal nodes exhibit a similar level of homophily, and (2) the labeled normal nodes can well represent the overall homophily pattern of the entire graph [6, 7, 12, 38, 42]. However, these two assumptions often do not hold well in the real-world GAD datasets. This is because although normal nodes generally exhibit high homophily, there can be large variations of homophily across the normal nodes, some of which can possess relatively low homophily (see Fig. 1**a** for visualization of this observation on the

---

[*]Equal Contribution. The work was done when Guoguo Ai visited Singapore Management University.
[†]Corresponding Authors.

39th Conference on Neural Information Processing Systems (NeurIPS 2025).

Amazon [10] and Elliptic [43]. Additional visualizations can be found in App. D.1). As a result, the GAD methods that rely on the aforementioned assumptions learn inaccurate homophily patterns of the normal class, leading to the misclassificaiton of low-homophily normal nodes as abnormal. This is particularly true when they use the popular neighborhood aggregation mechanisms that are prone to produce oversmooth homophily representations [37].

To address this issue, we propose **RHO**, namely Robust Homophily Learning, to adaptively learn heterogeneous normal patterns from the small set of normal nodes with diverse homophily.

RHO consists of two novel modules, including adaptive freqency response filters (**AdaFreq**) and graph normality alignment (**GNA**). Unlike existing filters that are bounded by prior knowledge of a specific homophily, AdaFreq learns a set of adaptive filters with learnable parameters to robustly fit normal nodes of varying levels of homophily. For example, as illustrated in Fig. 1**b** and 1**c**, conventional GCN filters [42] rely on a low-frequency assumption of normal nodes, while the spectral filters in BWGNN [41] learns a pre-defined type of frequency response, both of which fail to learn generalized representations for normal nodes across three different homophily variations; AdaFreq is instead optimized with learnable parameters to fit adaptively to the normal nodes with three different homophily cases, showing much more robustness than the other two methods. In RHO, AdaFreq is utilized to learn such adaptive filters in both channel-wise and cross-channel views of node attributes, enabling the learning of heterogeneous normal patterns in diverse feature channels of the labeled normal nodes.

The set of filters learned in AdaFreq can differ from each other substantially. To obtain robust yet consistent representations of the normal nodes, the GNA module is introduced to enforce consistency of homophily representations learned across the filters. To this end, GNA constructs positive pairs using the representations from the respective channel-wise and

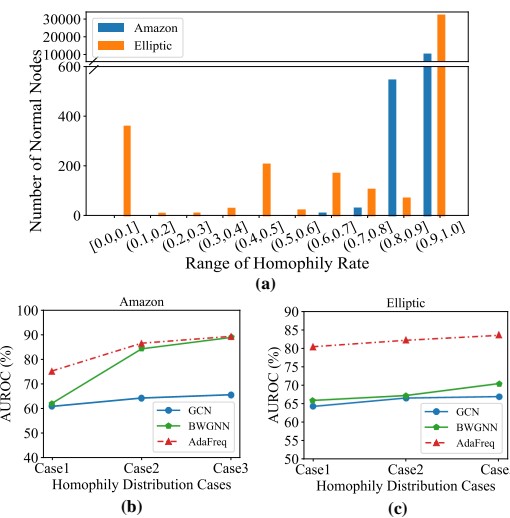

Figure 1: **(a)** Homophily distribution of normal nodes in Amazon [10] and Elliptic [43]. **(b)** and **(c)** AUC comparison of GCN filter [42], BWGNN filter [41], and AdaFreq filter on the two datasets under three cases of homophily levels. The three cases are normal nodes sampled from the sets of (low-homophily, high-homophily) normal nodes, where nodes with homophily greater than 0.9 for Amazon and 0.7 for Elliptic are considered as high homophily. The datasets in the three cases are sampled using a ratio of (80%, 20%), (50%, 50%), and (20%, 80%) to the two node sets, respectively.

cross-channel views at the corresponding nodes, and then maximizes the similarity between the representations learned from these two views while minimizing the similarity of negative pairs composed by representations from different nodes.

In doing so, RHO learns heterogeneous normal representations of normal nodes with varying degrees of homophily via AdaFreq, while ensuring the consistency of the learned normality via GNA. In summary, this work makes the following three main contributions.

- We reveal the failure of fitting existing graph filters to a subset of normal nodes with varying levels of homophily and introduce RHO to mitigate this issue. RHO is a novel approach for the semi-supervised GAD that can learn robust and consistent normal patterns for a given set of labeled normal nodes with diverse levels of homophily.

- We further introduce two novel modules, AdaFreq and GNA, to implement RHO. AdaFreq learns a set of complementary adaptive filters over channel-wise and cross-channel representations to effectively capture heterogeneous normal patterns from the labeled normal nodes of different levels of homophily. GNA robustifies these representations by aligning the normality representations learned by the filters in channel-wise and cross-channel views.

- Extensive experiments on eight real-world GAD datasets indicate that RHO performs significantly better than the state-of-the-art (SotA) semi-supervised GAD methods.

## 2 Related Work

### 2.1 Graph Anomaly Detection (GAD)

**Non-spectral GAD Methods.** Many non-sepctral GAD methods apply traditional anomaly detection techniques on a GNN backbone, such as reconstruction [7, 12], adversarial learning [5, 6], one-class classification [2, 42, 48]. Other methods are based on measures designed for GAD, such as the recently proposed affinity-based methods [30–32, 36]. These methods are typically designed for the unsupervised setting, where no labeled nodes are available. They can be easily adapted to the semi-supervised setting by refining their objective to the labeled normal nodes. GGAD [38] is the first method specifically designed for the semi-supervised setting by generating the outliers and training a discriminative one-class classifier with the given labeled normal nodes. These methods failed to consider the homophily discrepancy among the normal nodes. Our proposed RHO addresses this by employing an adaptive filter that learns normal representations from both cross-channel and channel-wise views through a one-class optimization objective.

**Spectral GAD Methods.** The spectral GAD methods focus on the filter design in GNNs to have a more effective learning the representations of nodes [3, 13, 40]. Among them, AMNet [3], BWGNN [40], and GHRN [13] aim to utilize the frequency signal using different spectral-based GNNs to distinguish between normal and abnormal nodes. Although these spectral-based GAD methods have achieved remarkable success [9, 24], they are typically designed for fully supervised setting where both labeled normal and abnormal nodes are available during training. On the other hand, the filters they utilize learn only predefined frequency information, making them difficult to learn expressive representations for normal nodes of different frequency components. RHO leverages AdaFreq to assign learnable parameters on different feature channels to effectively learn heterogeneous normal representations from the set of normal nodes of varying homophily.

### 2.2 Graph Homophily Modeling for GAD

Graph homophily modeling methods aim to address the homophily discrepancy in graphs, where target nodes often connect to nodes from different classes [14, 26, 45]. This inter-class connectivity negatively affects the quality of node representations within each class. The problem becomes more pronounced in GAD, where the homophily gap is typically large due to the inherent imbalance in GAD datasets [25, 37, 45]. Existing studies on Graph homophily modeling for GAD primarily focus on neighbor selection during message propagation by adding or cutting edges, or by assigning different weights to each edge [10, 21, 36]. In addition, several methods employ advanced strategies to address the homophily discrepancy in message passing, such as gradient-based filtering [13], meta-learning [8, 39], and data augmentation [4, 29, 47]. These methods overlook the homophily differences among the labeled normal nodes in semi-supervised settings. our proposed RHO is the first work to reveal this issue and address it by a robust homophily learning approach.

## 3 Problem Statement

**Notations.** Given an attributed graph $\mathcal{G} = (\mathcal{V}, \mathcal{E}, \mathbf{X})$, where $\mathcal{V}$ denotes the node set with $v_i \in \mathcal{V}$ and $|\mathcal{V}| = N$, $\mathcal{E}$ denotes the edge set, and $\mathbf{X} = [\mathbf{x}_1, \mathbf{x}_2, \ldots, \mathbf{x}_N] \in \mathbb{R}^{N \times M}$ is a set of node attributes. Each node $v_i$ has a $M$-dimensional feature representation $\mathbf{x}_i$. The topological structure of $\mathcal{G}$ is represented by an adjacency matrix $\mathbf{A}$. $\mathbf{D}$ denotes a degree matrix which is a diagonal matrix with $\mathbf{D}_{ii} = \sum_j \mathbf{A}_{ij}$. Normalized Laplacian matrix $\mathbf{L}$ is defined by $\mathbf{L} = \mathbf{I}_N - \mathbf{D}^{-\frac{1}{2}} \mathbf{A} \mathbf{D}^{-\frac{1}{2}}$, where $\mathbf{I}_N \in \mathbb{R}^{N \times N}$ denotes an identity matrix. $\hat{\mathbf{A}}$ is the normalized adjacency matrix. The corresponding degree matrix $\hat{\mathbf{D}}$ and Laplacian matrix $\hat{\mathbf{L}}$ are defined as $\hat{\mathbf{D}} = \mathbf{D} + \mathbf{I}_N$ and $\hat{\mathbf{L}} = \mathbf{I}_N - \hat{\mathbf{D}}^{-1/2} \hat{\mathbf{A}} \hat{\mathbf{D}}^{-1/2}$, respectively. The node homophily [34] for a node $v$ defined as $\mathcal{H}_v = \frac{|\{u | u \in \mathcal{N}_v, y_u = y_v\}|}{d_v}$, where $d_v$ is the number of neighbors of node $v$, $\mathcal{N}_v$ is the set of adjacent nodes of $v$, and $y_v$ represents the label of node $v$.

**Semi-supervised GAD.** Let $\mathcal{V}_a$, $\mathcal{V}_n$ be two disjoint subsets of $\mathcal{V}$, where $\mathcal{V}_a$ represents abnormal node set and $\mathcal{V}_n$ represents normal node set, and typically the number of normal nodes is significantly greater than the abnormal nodes, *i.e.*, $|\mathcal{V}_n| \gg |\mathcal{V}_a|$, then the goal of semi-supervised GAD is to learn the mapping function $\phi \to \mathbb{R}$, such that $\phi(v) < \phi(v')$, where $\forall v \in \mathcal{V}_n, v' \in \mathcal{V}_a$ , given a set of

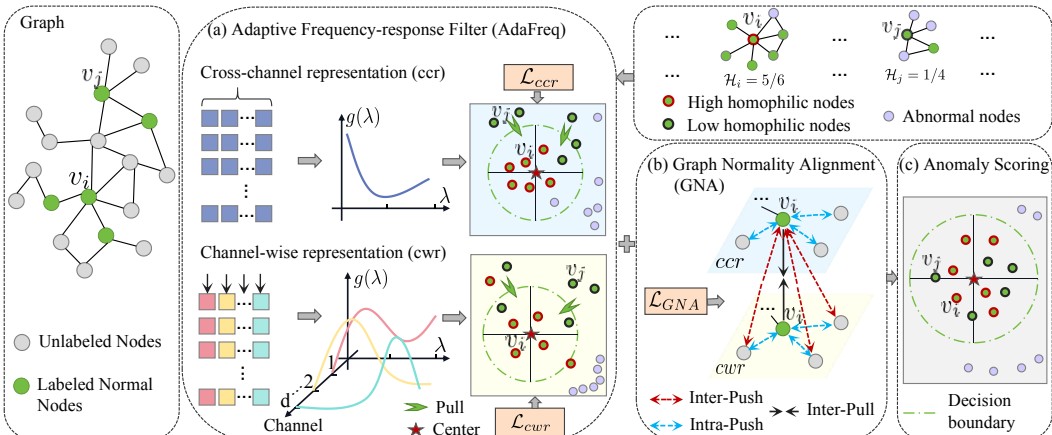

Figure 2: Overview of the proposed RHO framework. The input graph consists of labeled normal nodes and unlabeled normal/abnormal nodes, where nodes $v_i, v_j \in \mathcal{V}_l$ represent normal nodes with high and low homophily, respectively. **(a)** AdaFreq learns adaptive filters on both cross-channel and channel-wise representations with learnable parameters to different feature channels. **(b)** GNA aligns the heterogeneous normal patterns learned from the adaptive filters in the two views. **(c)** The normal nodes with diverse homophily are enforced to project closer to the center of a hypersphere via a widely-used one-class loss, while anomaly nodes being distant from the center.

labeled normal nodes $\mathcal{V}_l \subset \mathcal{V}_n$ and no access to any labels of the abnormal nodes. $\mathcal{V}_u = \mathcal{V}/\mathcal{V}_l$ is the set of the unlabeled nodes and used as test data.

**Graph Filter.** The Laplacian matrix $\hat{\mathbf{L}}$ can be decomposed as $\hat{\mathbf{L}} = \mathbf{U}\Lambda\mathbf{U}^\top$, where $\mathbf{U} = (\mathbf{u}_1, \mathbf{u}_2, \ldots, \mathbf{u}_N)$ is a complete set of orthonormal eigenvectors known as graph Fourier modes and $\Lambda = \mathrm{diag}\left(\{\lambda_i\}_{i=1}^N\right)$ is a diagonal matrix of the eigenvalues of $\hat{\mathbf{L}}$, where $\lambda_1 \leq \lambda_2 \leq \cdots \leq \lambda_N$ and $\lambda_i \in [0, 1.5]$ [44]. Taking the eigenvectors of normalized Laplacian matrix as a set of bases, graph Fourier transform of a signal $\mathbf{x} \in \mathbb{R}^N$ on graph $\mathcal{G}$ is defined as $\hat{\mathbf{x}} = \{\hat{x}_1, \cdots, \hat{x}_N\} = \mathbf{U}^T\mathbf{x}$, and the inverse graph Fourier transform is $\mathbf{x} = \mathbf{U}\hat{\mathbf{x}}$. The convolution between the signal $\mathbf{x}$ and convolution kernel $f$ is as follows:

$$f * \mathbf{x} = \mathbf{U}\left(\left(\mathbf{U}^T f\right) \odot \left(\mathbf{U}^T \mathbf{x}\right)\right) = \mathbf{U}g_\theta\mathbf{U}^T\mathbf{x}, \tag{1}$$

where $\odot$ is an element-wise product and $g_\theta$ is a diagonal matrix representing the convolution kernel in the spectral domain, serving as a substitute of $\mathbf{U}^T f$.

## 4 Methodology

### 4.1 Overview of the Proposed Approach RHO

As shown in Fig. 2, RHO consists of three main components: (1) Adaptive frequency response (AdaFreq) filters for heterogeneous normal pattern learning, which is simultaneously applies to both cross-channel and channel-wise view; (2) Graph normality alignment (GNA) for aligning the learned normality from two views using a contrastive learning objective; and (3) the model is lastly optimized with a widely-used one-class objective, along with a contrastive loss in GNA, to learn robust homophily patterns on GAD datasets with diverse levels of homophily. We also summarize the workflow of RHO and provide a detailed algorithmic description in App. E.

### 4.2 AdaFreq: Adaptive Frequency-response Filters

The conventional graph filter used in existing GAD methods is typically fixed, which fails to capture the diverse homophily relations of normal nodes. To this end, we are dedicated to learning adaptive frequency response filters that dynamically adjusts the response strength of different frequency components, effectively preserving the consistent frequency components of labeled normal nodes with diverse homophily. A straightforward strategy to control the varying effects of different frequency

components is to allocate a learnable parameter for each frequency component. However, this solution requires explicit eigen-decomposition which is too expensive. To avoid eigen-decomposition of the graph Laplacian, we propose a simple yet effective filter by introducing a trainable parameter $k$ to adjust the frequency response, as follows:

$$g(\lambda) = 1 - k\lambda, \tag{2}$$

where the learnable parameter $k$ governs the degree and type of frequency response imposed by the filter at each layer of GNNs. We show theoretically below that this design allows our filter $g(\lambda)$ to adaptively capture the consistent frequency patterns of normal nodes, regardless of having low- or high-homophily in the normal nodes.

**Theorem 1** *Let $\{\lambda_m\}$ and $\{\mathbf{u}_m\}$ be the graph frequencies and frequency components respectively, $\beta_m$ is the projection coefficient of signal $\mathbf{x}$ onto the $m$-th eigenvector $\mathbf{u}_m$, then we have $g(\lambda_m) = \frac{\sum_{i \in \mathcal{V}_l} u_m(i)}{\beta_m \sum_{i \in \mathcal{V}_l} u_m(i)^2}$ for the filter $g(\lambda)$, indicating that frequencies where normal nodes show coherent spectral behavior (i.e., $u_m(i)$ values agree in sign/magnitude) are amplified, while inconsistent frequency components are suppressed.*

The proof can be found in App. A. According to Theorem 1, the designed adaptive filter $g(\lambda)$ can preserve the frequency components with consistent spectral behavior of labeled normal nodes by training the parameter $k$. When $k > 0$, $g(\lambda)$ decreases with respect to $\lambda$, meaning that frequency components associated with larger eigenvalues (i.e., high-frequency components) are suppressed, while low-frequency components are preserved. Conversely, when $k < 0$, $g(\lambda)$ emphasizes high-frequency components. The magnitude of $k$ controls the strength of frequency responses. When $k = 0$, the filter $g(\lambda) = 1$ acts as an all-pass filter preserving the raw graph signal. When $K$ layers are stacked, $g(\lambda) = \prod_{i=1}^{K}(1 - k_i\lambda)$, a set of trainable parameters $\{k_1, k_2, \cdots, k_K\}$ can produce more complex frequency responses. Therefore, $g(\lambda)$ can dynamically learn the parameter $k$ to capturing the consistent frequency components of normal nodes with diverse homophily, enabling robust homophily representation learning for GAD.

Motivated by the intuition that heterogeneous homophily information is embedded in different feature channels, we leverage this adaptive frequency response function to learn heterogeneous patterns of the normal nodes in the cross-channel and channel-wise views as follows.

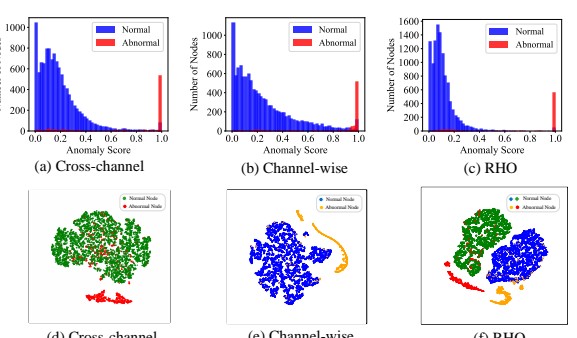

(a) Cross-channel  (b) Channel-wise  (c) RHO

(d) Cross-channel  (e) Channel-wise  (f) RHO

Figure 3: Anomaly score distributions and t-SNE visualizations of node embeddings generated by the cross-channel view (**(a)** and **(d)**), the channel-wise view (**(b)** and **(e)**), and the full RHO model (**(c)** and **(f)**) on Amazon [10].

**Homophily Learning across the Channels.** According to Eq. (1) and (2), AdaFreq can be utilized to learn a single learnable parameter $k \in \mathbb{R}$ shared across all node attributes in the cross-channel view. The learned **c**ross-**c**hannel **r**epresentations, denoted by $\mathbf{H}_{ccr}^{(t)}$ for the $t$-th layer, can be defined as:

$$\mathbf{H}_{ccr}^{(t)} = \sigma((\mathbf{I} - k\hat{\mathbf{L}})\mathbf{H}_{ccr}^{(t-1)}\mathbf{W}_{ccr}^{(t)}), \quad (3)$$

where $\sigma(\cdot)$ is a non-linear activation function, $\mathbf{H}_{ccr}^{(0)} = f_\theta(\mathbf{X})$ with $f_\theta$ be a two-layer MLP, and $\mathbf{W}_{ccr}^{(t)}$ is the learnable weight matrix.

To focus our filter on normal patterns, we map the representations of the normal nodes, $\mathbf{H}_{ccr}^{(T)}$, close to a one-class hypersphere center $\boldsymbol{c}_{ccr}$ in the embedding space. To achieve this, the training objective minimizes the average squared distance between the embeddings and the center:

$$\mathcal{L}_{ccr} = \frac{1}{|\mathcal{V}_l|} \sum_{i \in \mathcal{V}_l} \left\| \mathbf{h}_{ccr}^{(T)}(i) - \boldsymbol{c}_{ccr} \right\|^2 + \sum_{t=1}^{T} ||W_{ccr}^{(t)}||_F^2, \tag{4}$$

where $\mathbf{h}_{ccr}^{(T)}(i)$ denotes the representation of $i$-th node, *i.e.*, $i$-th row of $\mathbf{H}_{ccr}^{(T)}$, and $\boldsymbol{c}_{ccr} = \frac{1}{N} \sum_{v_i \in \mathcal{V}} \mathbf{h}_{ccr}^{(T)}(i)$. Optimizing $\mathcal{L}_{ccr}$ enables the filter to learn the heterogeneous normal patterns

from the cross-channel perspective, encouraging normal nodes of different homophily to cluster around the center $\boldsymbol{c}_{ccr}$, while anomalous nodes are distant from the center, as illustrated in Fig. 3a and the t-SNE visualization in Fig. 3d. However, relying solely on $\mathcal{L}_{ccr}$ may result in some anomalous nodes being located near the center, likely because they are camouflaged anomalies exhibiting distribution patterns similar to those of normal nodes in the cross-channel view.

**Homophily Learning in Individual Channels.** To capture normal patterns complementary to those in $\mathbf{H}_{ccr}^{(t)}$, we utilize AdaFreq in individual feature channels that can contain complementary homophily to the cross-channel view. To this end, we learn $d$ adaptive filters with each equipped with a learnable frequency response parameter, $\mathbf{K} = [k_1, k_2, ..k_d] \in \mathbb{R}^{1 \times d}$, where $d$ is the number of channels in the feature layer of the MLP network in $f_\theta$, to learn the **c**hannel-**w**ise **r**epresentations $\mathbf{H}_{cwr}^{(t)}$:

$$\mathbf{H}_{cwr}^{(t)} = \left[ \left( \mathbf{I} - k_1 \hat{\mathbf{L}} \right) \mathbf{h}_1^{(t-1)}, \left( \mathbf{I} - k_2 \hat{\mathbf{L}} \right) \mathbf{h}_2^{(t-1)}, \ldots, \left( \mathbf{I} - k_d \hat{\mathbf{L}} \right) \mathbf{h}_d^{(t-1)} \right] \mathbf{W}_{cwr}^{(t)}, \quad (5)$$

where $[\mathbf{h}_1^{(t-1)}, \mathbf{h}_2^{(t-1)}, \cdots, \mathbf{h}_d^{(t-1)}] = \mathbf{H}_{cwr}^{(t-1)} \in \mathbb{R}^{N \times d}$, and $\mathbf{H}_{cwr}^{(0)} = f_\theta(\mathbf{X})$.

These $d$ learnable frequency parameters allow the model to capture various homophily variations in our homophily pattern learning in individual channels. Alternatively, the filtering process can be compactly expressed using the Hadamard product as:

$$\mathbf{H}_{cwr}^{(t)} = \sigma((\mathbf{I} - \hat{\mathbf{L}})(\mathbf{H}_{cwr}^{(t-1)} \odot \mathbf{K}) \mathbf{W}_{cwr}^{(t)}). \quad (6)$$

The same one-class objective is applied to the channel-wise view, with its loss $\mathcal{L}_{cwr}$ defined as

$$\mathcal{L}_{cwr} = \frac{1}{|\mathcal{V}_l|} \sum_{i \in \mathcal{V}_l} \left\| \mathbf{h}_{cwr}^{(T)}(i) - \boldsymbol{c}_{cwr} \right\|^2 + \sum_{t=1}^{T} ||W_{cwr}^{(t)}||_F^2, \quad (7)$$

where $\mathbf{h}_{cwr}^{(T)}(i)$ is the $i$-th row of $\mathbf{H}_{cwr}^{(T)}$, $T$ is the number of layers, and $\boldsymbol{c}_{cwr} = \frac{1}{N} \sum_{v_i \in \mathcal{V}} \mathbf{h}_{cwr}^{(T)}(i)$. By optimizing $\mathcal{L}_{cwr}$, the model captures normal representation patterns that differ from those in the cross-channel view, as exemplified in Fig. 3b and Fig. 3e. However, placing too much emphasis on channel-wise features may cause certain normal nodes to drift away from the center, leading to some false positive detections.

Inspired by the complementary homophily information, AdaFreq learns the adaptive filters in both the cross-channel and channel-wise views to model the heterogeneous normal patterns. As shown in Fig. 3c and Fig. 3f, when we jointly leverage the two complementary views, normal nodes are more effectively clustered, and misclassified anomalies in Fig. 3d are successfully detected. Note that there are two centers since we apply AdaFreq to both views, and the GNA module we introduce in the next section is designed to align the heterogeneous normality represented by the two centers.

### 4.3 GNA: Graph Normality Alignment

There can be heterogeneous homophily representations learned in the two views above. We thus introduce GNA to mitigate this issue in AdaFreq. Specifically, for each node, we consider the representations generated by the adaptive filters as a positive pair, while representations from other nodes within the same batch are treated as negative pairs. GNA then promotes the normality alignment by maximizing the similarity of positive pairs and minimizing the similarity of negative pairs. Let $\mathbf{z}_{ccr}(i), \mathbf{z}_{cwr}(i) \in \mathbb{R}^d$ be the representations of node $v_i$ under the two views, which can be obtained using $\mathbf{z}_{ccr}(i) = \varphi_{ccr}(\mathbf{h}_{ccr}(i))$ and $\mathbf{z}_{cwr}(i) = \varphi_{cwr}(\mathbf{h}_{cwr}(i))$, where $\varphi_{ccr}(\cdot)$ and $\varphi_{cwr}(\cdot)$ are two MLP functions, then given the positive pair $(\mathbf{z}_{ccr}(i), \mathbf{z}_{cwr}(i))$, the alignment is formulated as

$$\mathcal{L}(\mathbf{z}_{ccr}(i), \mathbf{z}_{cwr}(i)) = -\log \frac{e^{(\text{sim}(\mathbf{z}_{ccr}(i), \mathbf{z}_{cwr}(i))/\tau)}}{\sum_{i \neq j} e^{(\text{sim}(\mathbf{z}_{ccr}(i), \mathbf{z}_{cwr}(j))/\tau)} + \sum_{i \neq j} e^{(\text{sim}(\mathbf{z}_{ccr}(i), \mathbf{z}_{ccr}(j))/\tau)}}, \quad (8)$$

where $\text{sim}(\cdot, \cdot)$ denotes cosine similarity, and $\tau$ is a temperature hyperparameter.

To better learn similarity of two views, we further designed an auxiliary loss module $\mathcal{L}(\mathbf{z}_{cwr}(i), \mathbf{z}_{ccr}(i))$ treating channel-wise view as anchor, contrasting to the $\mathcal{L}(\mathbf{z}_{cwr}(i), \mathbf{z}_{ccr}(i))$ that treats the cross-channel view as the anchor. Therefore, the overall objective of $\mathcal{L}_{GNA}$ is defined as

$$\mathcal{L}_{GNA} = \frac{1}{2N} \sum_{i=1}^{N} (\mathcal{L}(\mathbf{z}_{ccr}(i), \mathbf{z}_{cwr}(i)) + \mathcal{L}(\mathbf{z}_{cwr}(i), \mathbf{z}_{ccr}(i))). \quad (9)$$

By minimizing $\mathcal{L}_{GNA}$, the model is encouraged to capture consistent normality patterns across the cross-channel and channel-wise views, achieving robustness homophily representation learning, with better ability to detect the abnormal nodes, *e.g.*, the hard anomalies in Fig. 3c and 3f.

## 4.4 Training and Inference

**Training.** During training, RHO is guided by the one-class loss to learn heterogeneous normal patterns through the filters in AdaFreq and the alignment in GNA. To be specific, the total loss function $\mathcal{L}_{total}$ is formulated as a combination of the one-class classification loss, applied to the two views in AdaFreq, and the alignment loss $\mathcal{L}_{GNA}$ in GNA.

$$\mathcal{L}_{total} = \frac{1}{2}(\mathcal{L}_{ccr} + \mathcal{L}_{cwr}) + \alpha\mathcal{L}_{GNA}, \tag{10}$$

where $\alpha \in [0, 1]$ is a hyper-parameter to adjust the influence of $\mathcal{L}_{GNA}$ in the overall optimization.

**Inference.** During inference, the anomaly score of a node $v_i \in \mathcal{V}_u$ is defined as the squared Euclidean distance between the learned representations of a given node and the one-class center. In RHO, we compute the average distances of each node to both the cross-channel and channel-wise centers. The anomaly score $S_i$ for node $v_i$ is thus defined as

$$S_i = \frac{1}{2}(\left\|\mathbf{h}_{ccr}^{(T)}(i) - \mathbf{c}_{ccr}\right\|^2 + \left\|\mathbf{h}_{cwr}^{(T)}(i) - \mathbf{c}_{cwr}\right\|^2). \tag{11}$$

The abnormal nodes are located farther from the centers compared to the normal nodes and are therefore expected to have higher anomaly scores than the normal nodes.

## 4.5 Complexity Analysis

The computational complexity of RHO consists of three parts: (1) The two-layer MLP used for initial feature transformation has a complexity of $\mathcal{O}(Nd(M + d))$, where $N$ denotes the number of nodes, $M$ is the input feature dimension, and $d$ represents the hidden dimension. (2) RHO employs two independent views, including the cross-channel and channel-wise views. Each view incurs a computational cost of $\mathcal{O}(|\mathcal{E}|d + Nd^2)$, where $|\mathcal{E}|$ is the number of edges. The overall complexity of this component remains $\mathcal{O}(2 * (|\mathcal{E}|d + Nd^2))$ in practice. (3) In GNA, the projection head has a complexity of $\mathcal{O}(Nd^2)$, and the alignment loss computation has a complexity of $\mathcal{O}(Nbd)$, where $b$ denotes the batch size. Therefore, the overall time complexity is $\mathcal{O}(NMd + 2|\mathcal{E}|d + 4Nd^2 + Nbd)$. Empirical results for running time can be found in App. D.4.

## 5 Experiments

**Datasets.** We evaluate RHO on eight real-world GAD datasets of diverse size from different domains, including social networks Reddit [18] and Questions [35], co-review network Amazon [10], co-purchase network Photo [28], collaboration network Tolokers [28], financial networks T-Finance [41], Elliptic [43], and DGraph [15]. See App. B for more details about the datasets.

**Competing Methods.** The competing methods can be categorized into reconstruction methods (including DOMINANT [7] and AnomalyDAE [12]), one-class classification OCGNN [42], adversarial learning (including AEGIS [6] and GAAN [5]), affinity maximization method TAM [36], generative method GGAD [38], partitioning message passing method (PMP) [49], and data augmentation method CONSISGAD [4]. Except for GGAD that is specifically designed for the semi-supervised setting, the other methods are originally unsupervised or fully supervised GAD approaches, which are adapted to the semi-supervised scenario by refining the training set to the labeled normal nodes, following [38]. We also compare RHO against a conventional GCN filter and an advanced spectral-based filter used in BWGNN [41]. Note that for the fully supervised methods, we utilize only their proposed filter as the encoder and apply it within our semi-supervised setting by optimizing the encoder using the one-class loss. See App. C for more details about the competing methods. Note that some unsupervised methods, such as CoLA [23], SL-GAD [46], and GRADATE [11], are designed with proxy tasks and cannot be adapted to our setting, which are excluded from our comparison.

Table 1: AUROC and AUPRC on eight GAD datasets. The best performance is boldfaced, with the second-best underlined. '/' indicates that the model cannot handle DGraph.

| Metric | Methods | Datasets | | | | | | | |
|---|---|---|---|---|---|---|---|---|---|
| | | Reddit | Tolokers | Photo | Amazon | Elliptic | Question | T-Finance | DGraph |
| AUROC | DOMINANT | 0.5194 | 0.5121 | 0.5314 | 0.8867 | 0.3256 | 0.5454 | 0.6167 | 0.5851 |
| | AnomalyDAE | 0.5280 | 0.6074 | 0.5272 | 0.9171 | 0.5409 | 0.5347 | 0.6027 | 0.5866 |
| | OCGNN | 0.5622 | 0.4803 | 0.6461 | 0.8810 | 0.2881 | 0.5578 | 0.5742 | / |
| | AEGIS | 0.5605 | 0.4451 | 0.5936 | 0.7593 | 0.5132 | 0.5344 | 0.6728 | 0.4450 |
| | GAAN | 0.5349 | 0.3578 | 0.4355 | 0.6531 | 0.2724 | 0.4840 | 0.3636 | / |
| | TAM | 0.5829 | 0.4847 | 0.6013 | 0.8405 | 0.4150 | 0.5222 | 0.5923 | / |
| | GGAD | **0.6354** | 0.5340 | 0.6476 | **0.9443** | 0.7290 | 0.5122 | 0.8228 | 0.5943 |
| | GCN | 0.5523 | 0.4954 | 0.5727 | 0.7345 | 0.6463 | 0.4556 | 0.7262 | 0.5112 |
| | BWGNN | 0.5580 | 0.5821 | 0.6861 | 0.8312 | 0.7241 | 0.5740 | 0.7683 | 0.4958 |
| | PMP | 0.5472 | 0.5815 | 0.5844 | 0.8329 | 0.5617 | 0.5790 | 0.8321 | 0.5376 |
| | CONSISGAD | 0.5347 | 0.5974 | 0.5859 | 0.8715 | 0.7354 | 0.5737 | 0.8277 | 0.5735 |
| | RHO | 0.6207 | **0.6255** | **0.7129** | 0.9302 | **0.8509** | **0.5833** | **0.8623** | **0.6033** |
| AUPRC | DOMINANT | 0.0414 | 0.2217 | 0.1238 | 0.7289 | 0.0652 | 0.0314 | 0.0542 | 0.0076 |
| | AnomalyDAE | 0.0362 | 0.2697 | 0.1177 | 0.7748 | 0.0949 | 0.0317 | 0.0538 | 0.0071 |
| | OCGNN | 0.0400 | 0.2138 | 0.1501 | 0.7538 | 0.0640 | 0.0354 | 0.0492 | / |
| | AEGIS | 0.0441 | 0.1943 | 0.1110 | 0.2616 | 0.0912 | 0.0313 | 0.0685 | 0.0058 |
| | GAAN | 0.0362 | 0.1693 | 0.0768 | 0.0856 | 0.0611 | 0.0359 | 0.0324 | / |
| | TAM | 0.0446 | 0.2178 | 0.1087 | 0.5183 | 0.0552 | 0.0391 | 0.0551 | / |
| | GGAD | 0.0610 | 0.2502 | 0.1442 | **0.7922** | 0.2425 | 0.0349 | 0.1825 | **0.0082** |
| | GCN | 0.0420 | 0.2351 | 0.1257 | 0.1617 | 0.1176 | 0.0405 | 0.1387 | 0.0040 |
| | BWGNN | 0.0360 | 0.2687 | 0.2202 | 0.3797 | 0.1869 | 0.0410 | 0.1436 | 0.0041 |
| | PMP | 0.0388 | 0.2939 | 0.1254 | 0.3582 | 0.1006 | 0.0383 | 0.4084 | 0.0046 |
| | CONSISGAD | 0.0360 | 0.3059 | 0.1319 | 0.6523 | 0.1765 | 0.0424 | 0.4152 | 0.0050 |
| | RHO | **0.0616** | **0.3256** | **0.2337** | 0.7879 | **0.5095** | **0.0430** | **0.4893** | 0.0059 |

**Evaluation Metrics.** Following previous studies [19, 22, 38], we evaluate the detectors using two popular metrics: AUROC (Area Under the Receiver Operating Characteristic curve) and AUPRC (Area Under the Precision-Recall Curve). AUROC assesses the model's overall ability to distinguish between normal and anomalous nodes across varying thresholds. AUPRC measures the precision-recall tradeoff. For each method, we report the average results over five independent runs.

**Implementation Details.** The RHO model is implemented in PyTorch 2.0.0 with Python 3.8 and executed on GeForce RTX 3090 GPU (24 GB). RHO is trained using the Adam optimizer [16] with a weight decay of $5e^{-5}$. We set the default learning rate to $5e^{-3}$. Nevertheless, owing to the variations in edge density across different graphs, we find that models trained on sparser graphs are generally more sensitive to large learning rates and thus benefit from smaller ones to ensure stable convergence. Therefore, we use a learning rate of $5e^{-4}$ for the Elliptic and Question datasets, and further decreased to $5e^{-6}$ for the extremely sparse dataset, DGraph. The hyperparameter $\alpha$ is set to $1.0$ for all datasets except the small datasets Reddit and Photo, which require less regularization. It is set to $0.1$ in these two datasets. A detailed analysis of this hyperparameter is presented in Sec. 5.4. Following previous work [38], we randomly sample $R\%$ of the normal nodes as labeled data for training, where $R \in \{5, 10, 15, 20\}$. To ensure a fair comparison, we obtain the publicly-available official source code of all competitors and execute these models using the parameter settings suggested by their authors.

## 5.1 Main Comparison Results

The main comparison results are shown in Table 1, where all models use 15% labeled normal nodes during training. RHO outperforms the semi-supervised methods on six datasets having maximally 12.19% AUROC and 30.68% AUPRC improvement over the best competing method GGAD. GGAD achieves the highest AUROC on Amazon and Reddit; however, it underperforms RHO on the other datasets. This suggests that generative approaches may not generalize well across different datasets where normal nodes have varying levels of homophily. In contrast, RHO achieves SotA performance on a variety of graph datasets without relying on any generated anomalies, by learning the varying normal patterns from different datasets. The fully supervised methods PMP and CONSISGAD

perform less effectively when applied in the semi-supervised setting, since the lack of labeled anomalies prevents them from leveraging crucial abnormality information. The reconstruction-based method, AnomalyDAE, yields good performance on Tolokers and Amazon, but it still underperforms RHO on most datasets. The main reason is that reconstruction-based methods are prone to overfitting part of the prevalent homophily patterns (*e.g.*, low-frequency nodes) and struggle to handle hard normal samples (*e.g.*, those that have high homophily). In contrast, RHO leverages AdaFreq to effectively learn robust normal patterns from the normal nodes with diverse homophily, leading to substantially improved performance. As for one-class methods, OCGNN is based on non-adaptive filters in learning the normal patterns; it underperforms RHO across all datasets. Moreover, RHO also consistently outperforms GCN and BWGNN under the semi-supervised setting due to its superior capability in learning heterogeneous frequency components.

## 5.2 Analysis of Adaptive Filters in AdaFreq

We perform a filter analysis to examine the contribution of different frequency components in RHO. In Fig. 4, we provide a qualitative comparison between the filter in GCN and the $d+1$ adaptive filters learned by RHO in the two views. We observe that: (i) AdaFreq in cross-channel view flexibly retains varying levels of low-frequency signals across two datasets with diverse homophily patterns in the normal nodes, thereby enabling more effective learning of the heterogeneous normal representations. (ii) The channel-wise learnable filters, on the other hand, retain or enhance the discriminability of the frequency components at different levels, allowing RHO to capture

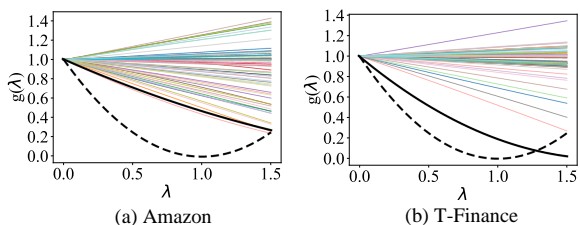

(a) Amazon      (b) T-Finance

Figure 4: The filter curves learned by RHO on Amazon and T-Finance, where the *black dashed line* represents the baseline filter response from GCN, the *black solid line* indicates the cross-channel response, and the *colored solid lines* are the channel-wise responses.

important homophily patterns specific to each feature channel. These two capabilities sharply contrast to the simple, fixed frequency pattern in GCN and its uniform treatment of all feature channels.

## 5.3 Ablation Study

In this section, we perform ablation study to evaluate the contribution of each component in RHO.

Table 2: AUROC and AUPRC results of our ablation study.

| Metric | Component | | | Datasets | | | | | | | |
|--------|---------------------|---------------------|---------------------|--------|----------|--------|--------|----------|----------|-----------|--------|
| | $\mathcal{L}_{ccr}$ | $\mathcal{L}_{cwr}$ | $\mathcal{L}_{GNA}$ | Reddit | Tolokers | Photo | Amazon | Elliptic | Question | T-Finance | DGraph |
| AUROC | ✓ | | | 0.5756 | 0.5712 | 0.6322 | 0.7436 | 0.7371 | 0.5704 | 0.7682 | 0.5542 |
| | | ✓ | | 0.6095 | 0.5878 | 0.6144 | 0.8905 | 0.7718 | 0.5811 | 0.8123 | 0.5215 |
| | ✓ | ✓ | | 0.6117 | 0.5727 | 0.6361 | 0.8692 | 0.7954 | 0.5774 | 0.7756 | 0.5637 |
| | ✓ | ✓ | ✓ | **0.6207** | **0.6255** | **0.7129** | **0.9302** | **0.8509** | **0.5833** | **0.8623** | **0.6033** |
| AUPRC | ✓ | | | 0.0431 | 0.3052 | 0.1390 | 0.2423 | 0.1862 | 0.0424 | 0.3282 | 0.0049 |
| | | ✓ | | 0.0481 | 0.2923 | 0.1451 | 0.5895 | 0.2287 | 0.0420 | **0.5061** | 0.0043 |
| | ✓ | ✓ | | 0.0556 | 0.3024 | 0.1398 | 0.5443 | 0.2657 | 0.0426 | 0.3365 | 0.0051 |
| | ✓ | ✓ | ✓ | **0.0616** | **0.3256** | **0.2337** | **0.7879** | **0.5095** | **0.0430** | 0.4893 | **0.0059** |

**Effectiveness of GNA.** To evaluate the effectiveness of GNA in RHO, we assess the variant of RHO with $\mathcal{L}_{GNA}$ removed and train the model using only $\mathcal{L}_{ccr}$ and $\mathcal{L}_{cwr}$. As shown in Table 2, removing $\mathcal{L}_{GNA}$ consistently decreases performance in terms of both AUROC and AUPRC. This indicates that $\mathcal{L}_{GNA}$ is crucial for bridging the gaps among the heterogeneous normal representations learned by the channel-wise and cross-channel views.

**Effectiveness of AdaFreq.** To verify the effectiveness of the proposed AdaFreq, we present the results of using AdaFreq in only one of the two views (*i.e.*, using solely $\mathcal{L}_{ccr}$ or $\mathcal{L}_{cwr}$). Table 2 shows that AdaFreq applied to both views consistently outperforms the variant that applies AdaFreq in only one of these views. This performance improvement is primarily attributed to the complementary nature of the two views. The homophily patterns captured in the channel-wise view contain distinct

information that cannot be fully learned from the cross-channel view alone, as exemplified in Fig. 4. The full model of RHO effectively integrates both views, resulting in improved performance across the datasets.

## 5.4 Hyperparameter Sensitivity Analysis

We evaluate the sensitivity of RHO w.r.t. the hyperparameter $\alpha$ and the training size $R$. Detailed results on more datasets can be found in App. D.

**Performance w.r.t. Hyperparameter $\alpha$.**

As shown in Fig. 5, with increasing $\alpha$, the performance on Elliptic, Tolokers, Amazon, Question and T-Finance has different degrees of improvement, suggesting that a stronger consistency constraint between cross-channel and channel-wise representations is beneficial. In contrast, on Reddit and Photo, the performance slightly declines as $\alpha$ increases. This is primarily because these are small datasets with high average feature similarity among the nodes. The representations learned from the two views are already highly similar, and imposing excessive regularization through the normality alignment may lead to overly smoothing features, which may lead to negative effects.

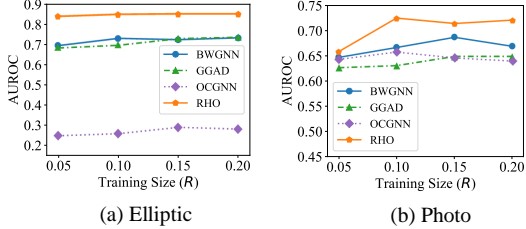

Figure 5: AUROC and AUPRC results of RHO w.r.t. hyperparamer $\alpha$.

**Performance w.r.t. Training Size $R$.**

To explore the impact of training size $R$, we compare RHO with GGAD, OCGNN, and BWGNN, using varying numbers of training normal nodes, with the results reported in Fig. 6. As the number of the labeled normal nodes increases, RHO generally improves across the datasets, which is consistent with other semi-supervised methods. Notably, RHO demonstrates a stable and remarkable improvement trend on the Elliptic dataset, even when the amount of labeled data is limited. This observation indicates that our model can effectively exploit the normality patterns under sparse supervision. Under different label rates, our RHO consistently maintains the best performance, showing the superiority of our method.

Figure 6: AUROC w.r.t. data size $R$.

## 6 Conclusion and Future Work

In this paper, we propose RHO, the very first GAD approach designed to learn heterogeneous normal patterns on a set of labeled normal nodes. RHO is implemented by two novel modules, AdaFreq and GNA. AdaFreq learns a set of adaptive spectral filters in both the cross-channel and channel-wise view of node attribute to capture the heterogeneous normal patterns from the given limited labeled normal nodes, while GNA is designed to enforce the consistency of the learned normal patterns, thereby facilitating the learning of robust normal representations on datasets with different levels of homophily in the normal nodes. This robustness is comprehensively verified by results on eight GAD datasets. A potential limitation of RHO is that it is a transductive method and thus cannot be directly applied in inductive settings. This limitation is shared with most existing GAD methods and is left for future exploration.

## Acknowledgments

This research is supported by the Ministry of Education, Singapore under its Tier-1 Academic Research Fund (24-SIS-SMU-008), A*STAR under its MTC YIRG Grant (M24N8c0103), and the Lee Kong Chian Fellowship.

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

# A Theoretical Analysis

A graph filter operation on a graph signal $\mathbf{x} \in \mathbb{R}^N$ can be defined as $\mathbf{z} = \mathbf{U}g(\mathbf{\Lambda})\mathbf{U}^\top\mathbf{x}$, where $\mathbf{U} = (\mathbf{u}_0, \mathbf{u}_1, \ldots, \mathbf{u}_{N-1})$ is a complete set of orthonormal eigenvectors of the Laplacian matrix, $g(\mathbf{\Lambda})$ is an adaptive filter with a diagonal matrix and $g(\lambda_0), g(\lambda_1), \cdots, g(\lambda_{N-1})$ is its main diagonal. In graph signal processing (GSP), $\{\lambda_i\}$ and $\{\mathbf{u}_i\}$ are called frequencies and frequency components of graph $\mathcal{G}$. Let $\mathcal{V}_l$ be the labeled normal nodes in our training data and $c$ be a center of a one-class classifier, then the objective function of the one-class classification under semi-supervised graph anomaly detection (GAD) can be defined as:

**Definition A.1** *(Spectral One-class Loss) Let $\boldsymbol{\beta} = \mathbf{U}^\top\mathbf{x}$, then in the one-class classification scenario, the loss of filter $g(\mathbf{\Lambda})$ on a graph can be formulated as follows:*

$$\mathcal{L} = \sum_{i \in \mathcal{V}_l} ||(\mathbf{U}g(\mathbf{\Lambda})\boldsymbol{\beta})_i - c||^2 = \sum_{i \in \mathcal{V}_l} ||z_i - c||^2, \tag{12}$$

where $\boldsymbol{\beta} = (\beta_0, \beta_1, \cdots, \beta_{N-1})^\top$, with $\beta_m$ be the projection coefficient of $\mathbf{x}$ on the $m$-th eigenvector $\mathbf{u}_m$ (i.e., $\mathbf{x} = \sum_{m=0}^{N-1} \beta_m \mathbf{u}_m$), and the adaptive filter $g(\mathbf{\Lambda})$ operates on each spectral component, i.e., $g(\mathbf{\Lambda})\boldsymbol{\beta} = (g(\lambda_0)\beta_0, g(\lambda_1)\beta_1, \cdots, g(\lambda_{N-1})\beta_{N-1})$. Then, through the inverse transformation, we obtain the node representation $\mathbf{z} = \sum_{m=0}^{N-1}(g(\lambda_m)\beta_m)\mathbf{u}_m$, with $z_i = \sum_{m=0}^{N-1}(g(\lambda_m)\beta_m)u_m(i)$, where $u_m(i)$ is the value of the eigenvector $\mathbf{u}_m$ at node $i$.

**Theorem 2** *Let $\{\lambda_m\}$ and $\{\mathbf{u}_m\}$ be the graph frequencies and frequency components respectively, $\beta_m$ is the projection coefficient of signal $\mathbf{x}$ onto the $m$-th eigenvector $\mathbf{u}_m$, then $g(\lambda_m) = \frac{\sum_{i \in \mathcal{V}_l} u_m(i)}{\beta_m \sum_{i \in \mathcal{V}_l} u_m(i)^2}$ holds for the filter $g(\lambda)$, indicating that frequencies where normal nodes show coherent spectral behavior (i.e., $u_m(i)$ values agree in sign/magnitude) are amplified, while inconsistent frequency components are suppressed.*

**Proof 1** *Starting from the one-class classification loss over the labeled normal nodes $\mathcal{V}_l$:*

$$\mathcal{L} = \sum_{i \in \mathcal{V}_l} \left\| \sum_{m=0}^{N-1} g(\lambda_m)\beta_m u_m(i) - c \right\|^2.$$

*we compute the gradient of $\mathcal{L}$ with respect to $g(\lambda_m)$:*

$$\frac{\partial\mathcal{L}}{\partial g(\lambda_m)} = 2\sum_{i \in \mathcal{V}_l} \left( \sum_{j=0}^{N-1} g(\lambda_j)\beta_j u_j(i) - c \right) \cdot \beta_m u_m(i).$$

*Setting the derivative to zero for optimality and assuming orthogonality between eigenvectors (i.e., dropping cross terms for $m \neq j$), we obtain:*

$$g(\lambda_m)\beta_m \sum_{i \in \mathcal{V}_l} u_m(i)^2 = c \sum_{i \in \mathcal{V}_l} u_m(i).$$

*Solving for $g(\lambda_m)$ yields the closed-form by assuming the center $c = 1$:*

$$g(\lambda_m) = \frac{\sum_{i \in \mathcal{V}_l} u_m(i)}{\beta_m \sum_{i \in \mathcal{V}_l} u_m(i)^2}.$$

In the theorem, the denominator $\beta_m \sum_{i \in \mathcal{V}_l} u_m(i)^2$ represents the weighted total energy of the $m$-th frequency component distributed over the normal nodes. It serves as a normalization factor, reflecting how uniformly the $m$-th frequency component is distributed across the normal nodes. When the $m$-th frequency components exhibit high consistency across the labeled normal nodes, e.g., $u_m(i)$ have the same sign (all positive or all negative) and similar amplitude for $i \in \mathcal{V}_l$, then the ratio $\frac{\sum_{i \in \mathcal{V}_l} u_m(i)}{\beta_m \sum_{i \in \mathcal{V}_l} u_m(i)^2}$ will be either much greater than 0 (case 1: $\beta_m > 0$ and $\frac{\sum_{i \in \mathcal{V}_l} u_m(i)}{\sum_{i \in \mathcal{V}_l} u_m(i)^2} \gg 0$, case 2: $\beta_m < 0$ and $\frac{\sum_{i \in \mathcal{V}_l} u_m(i)}{\sum_{i \in \mathcal{V}_l} u_m(i)^2} \ll 0$) or much less than 0 (case 1: $\beta_m > 0$ and $\frac{\sum_{i \in \mathcal{V}_l} u_m(i)}{\sum_{i \in \mathcal{V}_l} u_m(i)^2} \ll 0$,

case 2: $\beta_m < 0$ and $\frac{\sum_{i\in\mathcal{V}_l} u_m(i)}{\sum_{i\in\mathcal{V}_l} u_m(i)^2} \gg 0$), resulting in a frequency positively or negatively enhancing response $g(\lambda_m) \gg 0$ or $g(\lambda_m) \ll 0$. Only when the frequency components are highly inconsistent, *e.g.*, $u_m(i)$ exhibits opposite signs across the labeled normal nodes, these inconsistent components will cancel each other out (*i.e.*, $\sum_{i\in\mathcal{V}_l} u_m(i) \to 0$), resulting in a frequency suppressing response $g(\lambda_m) \to 0$. To summarize, minimizing the one-class loss enables an adaptive filtering mechanism to automatically enhance frequency components that are consistent across normal nodes while suppressing the inconsistent components.

Table 3: Key statistics of GAD datasets.

| Datasets | Reddit | Tolokers | Photo | Amazon | Elliptic | Question | T-Finance | DGraph |
|---|---|---|---|---|---|---|---|---|
| #Nodes | 10,984 | 11,758 | 7,484 | 11,944 | 203,769 | 48,921 | 39,357 | 3,700,550 |
| #Edges | 168,016 | 519,000 | 119,043 | 4,398,392 | 234,355 | 153,540 | 21,222,543 | 4,300,999 |
| #Attributes | 64 | 10 | 745 | 25 | 166 | 301 | 10 | 17 |
| Anomaly | 3.3% | 21.8% | 4.9% | 9.5% | 9.8% | 2.98% | 4.6% | 1.3% |

# B  Detailed Description of Datasets

The key statistics of the datasets are presented in Table 3. A detailed introduction of these datasets is given as follows.

- Reddit [18]: It is a user-subreddit graph which captures one month's worth of posts shared across various subreddits at Reddit. The node represents the users, and the text of each post is transformed into a feature vector and the features of the user and subreddits are the feature summation of the post they have posted. The anomalies are the used who have been banned by the platform.

- Tolokers [28]: It is obtained from the Toloka crowdsourcing platform, where the node represents the person who has participated in the selected project, and the edge represents two workers work on the same task. The attributes of the node are the profile and task performance statistics of workers.

- Photo [28]: It is obtained from an Amazon co-purchase network where the node represents the product and the edge represents the co-purchase relationship. The bag-of-words representation of the user's comments is used as the attribute of the node.

- Amazon [10]: It is a co-review network obtained from the Musical Instrument category on Amazon.com. There are also three relations: U-P-U (users reviewing at least one same product), U-S-U (users having at least one same star rating within one week), and U-V-U (users with top-5% mutual review similarities).

- Elliptic [43]: It is a Bitcoin transaction network in which each node represents a transaction and an edge indicates a flow of Bitcoin currency. The Bitcoin transaction is mapped to real-world entities associated with licit categories in this dataset.

- Question [35]: It is a question answering network which is collected from the website Yandex Q where the node represents the user and the edge connecting the node represents the user who has answered other's questions during a one-year period. The attribute of nodes is the mean of FastText embeddings for words in the description of the user. For the user without a description, the additional binary features are employed as the feature of the user.

- T-Finance [41]: It is a financial transaction network where the node represents an anonymous account and the edge represents two accounts that have transaction records. Some attributes of logging like registration days, logging activities, and interaction frequency, etc, are used as the features of each account. Users are labeled as anomalies if they fall into categories such as fraud, money laundering, or online gambling.

- DGraph [15]: It is a large-scale attributed graph with millions of nodes and edges where the node represents a user account in a financial company and the edge represents that the user was added to another account as an emergency contact. The feature of each node represents the user's profile information, including age, gender, and other demographic attributes. Users with a history of overdue payments are labeled as anomalies.

## C   Description of Baselines

A more detailed introduction of the nine competing GAD models is given as follows.

- DOMINANT [7] applied the conventional autoencoder on a graph for GAD. It consists of an encoder layer and a decoder layer, which are designed to reconstruct the features and structure of the graph. The reconstruction errors from the features and the structural modules are combined as an anomaly score.

- AnomalyDAE [12] leverages the advanced autoencoder and an attribute autoencoder to learn both node embeddings and attribute embeddings jointly in a latent space by assigning the corresponding weight in the loss function. In addition, an attention mechanism is employed in the structure encoder to capture normal structural patterns more effectively.

- OCGNN [42] combines one-class SVM and GNNs, aiming at leveraging one-class classifiers and the powerful representation of GNNs. A one-class hypersphere learning objective is used to drive the training of the GNN. The samples that fall outside the hypersphere, meaning they deviate from the normal pattern, are defined as anomaly.

- AEGIS [6] designs a new graph neural layer to learn anomaly-aware node representations and further employ generative adversarial networks to detect anomalies among new data.The generator takes noise sampled from a prior distribution as input and aims to produce informative pseudo-anomalies. Meanwhile, the discriminator seeks to determine whether a given input is the representation of a normal node or a generated anomaly.

- GAAN [5] is based on a generative adversarial network where fake graph nodes are generated by a generator. To encode the nodes, they compute the sample covariance matrix for real nodes and fake nodes, and a discriminator is trained to recognize whether two connected nodes are from a real or fake node.

- TAM [36] adopts the local affinity-based approach as an anomaly measure. It learns tailored node representations for a new anomaly measure by maximizing the local affinity of nodes to their neighbors. TAM is optimized on truncated graphs where non-homophily edges are removed iteratively. The learned representations result in a significantly stronger local affinity for normal nodes than abnormal nodes.

- GGAD [38] generates pseudo anomaly nodes based on two important priors, including asymmetric local affinity and egocentric closeness, and trains a discriminative one-class classifier for semi-supervised GAD.

- GCN [17] obtains the node representations by aggregating information from neighbors, which can be seen as a special form of low-pass filter. The low-pass filter primarily preserves the similarity of node features during graph representation learning.

- BWGNN [41] reveals the 'right shift' phenomenon that the spectral energy distribution concentrates more on the node with high frequencies and less on the node with low frequencies, and proposes a band-pass filter to learn the effective node representation for supervised GAD.

- PMP [49] introduces a partitioning message passing scheme that separately aggregates information from homophilic and heterophilic neighbors using node-specific functions. This design enables the adaptive adjustment of information flow from different neighbor types, thereby effectively capturing structural heterogeneity and enhancing robustness to heterophily in graph-based fraud detection.

- CONSISGAD [4] introduces a consistency-based framework for GAD under limited supervision. It leverages unlabeled data through a learnable data augmentation mechanism that injects controlled noise for consistency training. Additionally, by exploiting the variance in homophily distributions between normal and anomalous nodes, CONSISGAD employs a simplified GNN backbone to enhance discriminability and robustness against class imbalance.

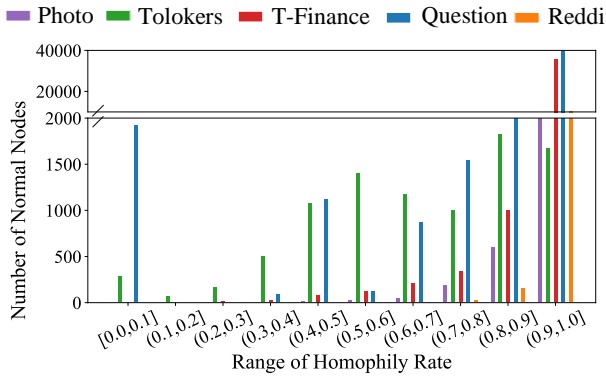

Figure 7: Homophily distributions of normal nodes in five datasets.

# D  Additional Experimental Results

## D.1  Homophily Distribution Results for the Remaining Datasets

In Fig. 7, we visualize the homophily distributions of normal nodes on the rest of five datasets. As shown in Fig. 1, the observation is consistent with the distributions on Amazon and Elliptic, *i.e.*, although most normal nodes exhibit high homophily, some normal nodes show relatively low homophily, resulting in significant variations in homophily levels among the normal nodes. This further indicates that this phenomenon is commonly observed in the GAD datasets, and existing models based on the aforementioned assumptions may learn inaccurate homophily patterns of the normal class on these datasets, while RHO adaptively learning heterogeneous normal patterns from the small set of normal nodes with diverse homophily can well address this challenge.

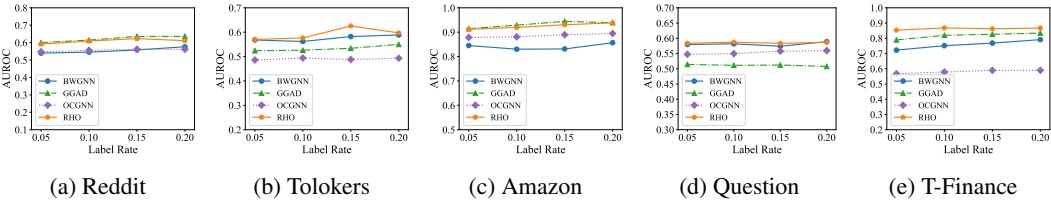

| (a) Reddit | (b) Tolokers | (c) Amazon | (d) Question | (e) T-Finance |

Figure 8: AUROC w.r.t. training data size $R$

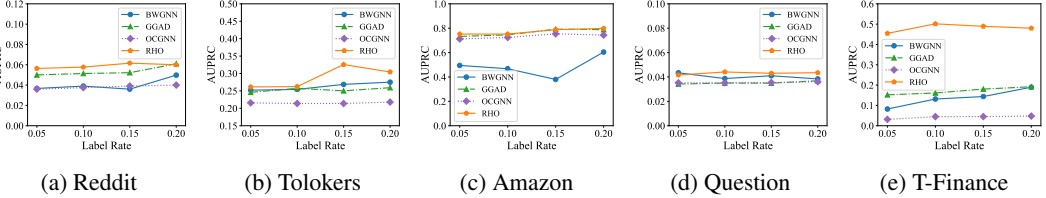

| (a) Reddit | (b) Tolokers | (c) Amazon | (d) Question | (e) T-Finance |

Figure 9: AUPRC w.r.t. training data size $R$

## D.2  Performance w.r.t. Different Training Size

The AUROC and AUPRC results under varying proportions of training normal nodes are shown in Fig. 8 and Fig. 9, respectively. The results further demonstrate that as the number of labeled normal nodes increases, the performance of RHO generally improves across the datasets, which is consistent with the behavior observed in other semi-supervised methods. Besides, RHO consistently outperforms the competing methods across different numbers of training normal nodes on Tolokers, Question, and T-Finance in terms of AUROC. RHO also achieves the best AUPRC performance across all datasets under varying training data sizes, further demonstrating its effectiveness in varying homophily pattern learning in the small normal node set.

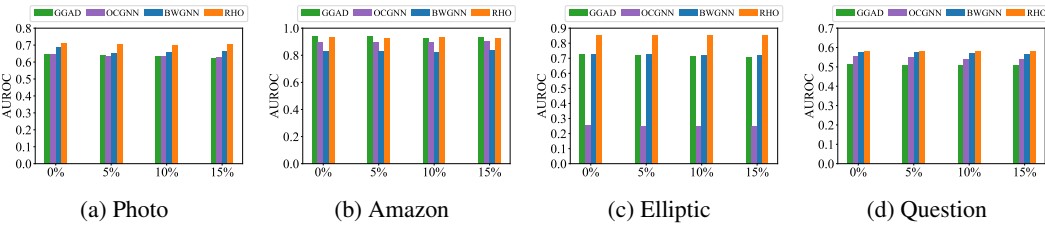

Figure 10: AUROC results w.r.t. different anomaly contamination rates.

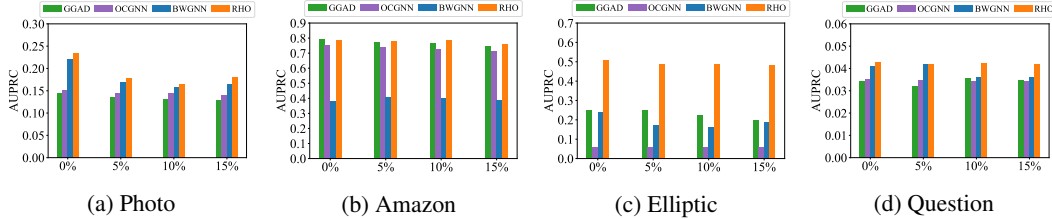

Figure 11: AUPRC results w.r.t. different anomaly contamination rates.

### D.3 Performance w.r.t. Anomaly Contamination

In real applications, labeled normal nodes are often susceptible to contamination by anomalies due to factors such as annotation errors. To address this issue, we introduce a controlled ratio of anomaly contamination into the training set of labeled normal nodes. Specifically, we randomly sample abnormal nodes from the unlabeled nodes in the test set and incorporate them as normal nodes into the training set of normal node set. All sampled abnormal nodes are excluded from the test set.

As shown in Fig. 10 and Fig. 11, the performance of all models declines as the level of contamination increases, particularly on the AUPRC. RHO achieves the best performance among all competing methods and maintains consistent results across varying contamination rates, demonstrating its robustness in representation learning for semi-supervised GAD.

### D.4 Runtime Results

The running time, including both training and inference time, of RHO and six competing methods are shown in Table 4. Although our method can run on all datasets using an RTX 3090 GPU, some baseline methods may encounter out-of-memory (OOM) issues on the same GPU due to their different computational mechanisms, highlighting that RHO is more memory-efficient. To ensure a fair comparison, we report the runtime

Table 4: Runtimes (in seconds) on the six datasets on CPU.

| Methods | Datasets | | | | |
|---|---|---|---|---|---|
| | Reddit | Photo | Amazon | Question | T-Finance |
| DoMINANT | 125 | 437 | 1592 | 740 | 10721 |
| OCGNN | 162 | 125 | 765 | 973 | 5717 |
| AEGIS | 166 | 417 | 1121 | 660 | 15258 |
| TAM | 432 | 165 | 4516 | 29280 | 17360 |
| GGAD | 368 | 136 | 1020 | 1125 | 9345 |
| CONSISGAD | 483 | 681 | 3259 | 483 | 20172 |
| RHO | 398 | 201 | 3358 | 675 | 5184 |

performance on the CPU for all methods. The runtime results demonstrate that RHO remains efficient even on the large-scale graph T-Finance, primarily due to the use of mini-batch processing in the GNA module. By adopting a mini-batch strategy, the complexity of GNA reduces to $O(b^2 d)$ for a batch size of $b$, and the total per-epoch complexity becomes $O(Nbd)$ when averaged across all batches, which significantly reduces memory usage and computational overhead, thereby improving the scalability. In contrast, methods such as DOMINANT, TAM, and GGAD involve reconstruction or affinity calculations across all nodes, which typically incur a time complexity of $O(N^2)$, causing significant computational overhead on large-scale graphs.

**Algorithm 1** RHO: Semi-supervised Graph Anomaly Detection via Robust Homophily Learning

**Input**: Graph $\mathcal{G} = (\mathcal{V}, \mathcal{E})$, node features $\mathbf{X}$, labeled normal nodes $\mathcal{V}_l$, unlabeled nodes $\mathcal{V}_u$, number of training epochs $E$, temperature $\tau$, trade-off weight $\alpha$, batch size $b$, propagation depth $T$

**Output**: Anomaly scores $S_i$ for each $v_i \in \mathcal{V}_u$

1: Initialize parameters $\theta$, $\{\mathbf{W}_{ccr}^{(t)}, \mathbf{W}_{cwr}^{(t)}\}_{t=1}^{T}$, frequency coefficients $k$ (shared in cross-channel), $\mathbf{K} = [k_1, \ldots, k_d]$ (channel-wise)
2: Set initial features: $\mathbf{H}_{ccr}^{(0)} \leftarrow f_\theta(\mathbf{X})$, $\mathbf{H}_{cwr}^{(0)} \leftarrow f_\theta(\mathbf{X})$; $\boldsymbol{c}_{ccr} = \mathbf{0}, \boldsymbol{c}_{cwr} = \mathbf{0}$
3: **for** $t = 1$ to $T$ **do**
4:     Update cross-channel view: $\mathbf{H}_{ccr}^{(t)} \leftarrow \sigma\left((\mathbf{I} - k\hat{\mathbf{L}})\mathbf{H}_{ccr}^{(t-1)}\mathbf{W}_{ccr}^{(t)}\right)$
5:     Update channel-wise view: $\mathbf{H}_{cwr}^{(t)} \leftarrow \sigma\left((\mathbf{I} - \hat{\mathbf{L}})(\mathbf{H}_{cwr}^{(t-1)} \odot \mathbf{K})\mathbf{W}_{cwr}^{(t)}\right)$
6: **end for**
7: Compute centers: $\boldsymbol{c}_{ccr} \leftarrow \frac{1}{|\mathcal{V}|}\sum_{i \in \mathcal{V}} \mathbf{h}_{ccr}^{(T)}(i)$, $\boldsymbol{c}_{cwr} \leftarrow \frac{1}{|\mathcal{V}|}\sum_{i \in \mathcal{V}} \mathbf{h}_{cwr}^{(T)}(i)$
8: Compute projections: $\mathbf{z}_{ccr} \leftarrow \varphi_{ccr}(\mathbf{h}_{ccr}^{(T)})$, $\mathbf{z}_{cwr} \leftarrow \varphi_{cwr}(\mathbf{h}_{cwr}^{(T)})$
9: **for** $epoch = 1$ to $E$ **do**
10:     Compute one-class losses $\mathcal{L}_{ccr}, \mathcal{L}_{cwr}$ in $v_i \in v_l$
11:     **if** $b > 0$ **then**
12:         Sample a batch $\mathcal{B} \subset \mathcal{V}$ of size $b$
13:     **else**
14:         Set $\mathcal{B} \leftarrow \mathcal{V}$
15:     **end if**
16:     **for** each $v_i \in \mathcal{B}$ **do**
17:         $\mathcal{L}_{GNA} \leftarrow \frac{1}{2|\mathcal{B}|}\sum_i (\mathcal{L}(\mathbf{z}_{ccr}(i), \mathbf{z}_{cwr}(i)) + \mathcal{L}(\mathbf{z}_{cwr}(i), \mathbf{z}_{ccr}(i)))$
18:     **end for**
19:     $\mathcal{L}_{total} \leftarrow \frac{1}{2}(\mathcal{L}_{ccr} + \mathcal{L}_{cwr}) + \alpha\mathcal{L}_{GNA}$
20:     $\boldsymbol{c}_{ccr} \leftarrow \frac{1}{|\mathcal{V}|}\sum_{i \in \mathcal{V}} \mathbf{h}_{ccr}^{(T)}(i)$, $\boldsymbol{c}_{cwr} \leftarrow \frac{1}{|\mathcal{V}|}\sum_{i \in \mathcal{V}} \mathbf{h}_{cwr}^{(T)}(i)$
21:     Update all parameters via backpropagation
22: **end for**
23: **for** each $v_i \in \mathcal{V}_u$ **do**
24:     Anomaly scoring: $S_i = \frac{1}{2}(\|\mathbf{h}_{ccr}^{(T)}(i) - \boldsymbol{c}_{ccr}\|^2 + \|\mathbf{h}_{cwr}^{(T)}(i) - \boldsymbol{c}_{cwr}\|^2)$
25: **end for**
26: **return** Anomaly scores $S_1, \ldots, S_{|\mathcal{V}_u|}$

# E  Algorithm

The algorithm of RHO is summarized in Algorithm 1.

