# OpenReview forum: "Semi-supervised Graph Anomaly Detection via Robust Homophily Learning"
_NeurIPS.cc/2025/Conference — NeurIPS 2025 poster_

### Official Review · Reviewer_tFPZ · 2025-07-01

**Clarity:** 3
**Significance:** 3
**Originality:** 3
**Rating:** 5
**Confidence:** 4

**Summary:**

This work proposes a novel framework, Robust Homophily Learning (RHO), for semi-supervised graph anomaly detection (GAD). RHO incorporates two novel designs: AdaFreq, which captures diverse frequency components of labeled normal nodes across channel-wise and cross-channel views of node attributes; and GNA, which enforces consistency between two views to robustly model node normality, accounting for varying levels of homophily among normal nodes. Experiments on eight real-world GAD datasets demonstrate the effectiveness of RHO.

**Questions:**

Please refer to Weaknesses.

**Ethical Concerns:**

["NO or VERY MINOR ethics concerns only"]

**Final Justification:**

The authors' responses have addressed my concerns and further improved the paper's quality. Therefore, I would like to increase my rating.

**Limitations:**

Please refer to Weaknesses.

**Paper Formatting Concerns:**

No formatting concerns.

**Quality:**

3

**Strengths And Weaknesses:**

**Strengths:**

1. The paper is well written and easy to follow.

2. The motivation for the proposed RHO is novel, and theoretical analysis supports the design of the adaptive filter. The visualization in Figure 3 is intuitive.

3. The experiments and theory are sufficient to demonstrate the effectiveness.

**Weaknesses and Questions:**

1. According to the algorithm design of RHO, the centers $c$ in the two views are dynamically updated during training, which differs from the static center design used in previous works. What is the motivation behind this dynamic update? If the centers are set as static, how would it affect the model's performance?

2. The description of symbols are unclear. What is the difference between the convolution kernel $g$ in Line 145 and convolution kernel $g\_\theta$ in Line 146.

3. Figure 1a presents the statistics of the homophily rate for normal nodes. The formula provided for computing node-level homophily (Lines 128–129) is more appropriate than using graph-level homophily.

4. RHO achieves significant improvements on some datasets, but demonstrates competitive performance on Reddit and Amazon. What could be the possible reasons behind this?

---

> ### Author Rebuttal · Authors · 2025-07-31
>
> We sincerely appreciate your constructive and positive comments on the presentation, visualization, and empirical justification.  Please see our response to your comments one by one below.
>
> >**Weakness #1:** According to the algorithm design of RHO, the centers in the two views are dynamically updated during training, which differs from the static center design used in previous works. What is the motivation behind this dynamic update? If the centers are set as static, how would it affect the model's performance?
>
> Thank you very much for your comment. The motivation of using dynamic centers $c$ stems from our observation that the frequency signals of normal nodes may shift during training due to model updates and evolving spectral representations. A static center $c$, which remains fixed throughout the training, would fail to capture the evolving distribution of spectral embeddings, leading to suboptimal performance. In contrast, dynamically updating the center enables the model to adaptively refine the target frequency region in response to changes in the spectral representations, resulting in more effective feature learning. The collaborative mechanism between the filter and the center allows the learnable spectral filter to more effectively preserve the consistent frequency components associated with normal nodes.
>
>
> To validate effectiveness, we add and evaluate the results of RHO-Static, a variant of RHO that uses a fixed center. The experimental results are shown in Tables A1 and A2 below. From the results, we can observe that RHO with a dynamic center achieves the best results on almost all datasets, which validates the effectiveness of dynamically adapting the center.
>
> ```
> Table A1. AUROC results of RHO and RHO-Static
> ```
> |Datasets|Reddit|Tolokers|Photo|Amazon|Elliptic|Question|T-Finance|
> |---|---|---|---|---|---|---|---|
> |RHO-Static|0.5676|0.5988|**0.7234**|0.9153|**0.8509**|0.5729|0.8208|
> |RHO|**0.6207**|**0.6255**|0.7129|**0.9302**|**0.8509**|**0.5833**|**0.8623**|
>
> ```
> Table A2. AUPRC results of RHO and RHO-Static
> ```
> |Datasets|Reddit|Tolokers|Photo|Amazon|Elliptic|Question|T-Finance|
> |---|---|---|---|---|---|---|---|
> |RHO-Static|0.0468|0.2865|0.2311|0.7341|0.5092|0.0393|0.3023|
> |RHO|**0.0616**|**0.3256**|**0.2337**|**0.7879**|**0.5095**|**0.0430**|**0.4893**|
>
>
> >**Weakness #2:** The description of symbols are unclear. What is the difference between the convolution kernel $g$ in Line 145 and the convolutional kernel $g_\theta$ in Line 146.
>
> Thank you for pointing this out. We apologize for the confusion caused by this notation. $g$ and $g_\theta$ denote convolution kernels in the spatial and spectral domains, respectively.
>
> To provide a clearer explanation, we would replace $g$ with $f$, and revised the description of Equation (1) as follows.
>
> *The convolution between the signal $x$ and convolution kernel $f$ is:
>     $ f *  \mathbf{x}=\mathbf{U}\left(\left(\mathbf{U}^{T} f\right) \odot\left(\mathbf{U}^{T} \mathbf{x}\right)\right)=\mathbf{U} g_{\theta} \mathbf{U}^{T} \mathbf{x}$,
> where $\odot$ is an element-wise product and $g_{\theta}$ is a diagonal matrix, which represents the convolutional kernel in the spectral domain, replacing $\mathbf{U}^{T} f$.*
>
> We will clarity this and add this revised description in the final version.
>
> >**Weakness #3:** Figure 1a presents the statistics of the homophily rate for normal nodes. The formula provided for computing node-level homophily (Lines 128–129) is more appropriate than using graph-level homophily.
>
> Thank you very much for your valuable suggestion.  We will provide the definition of the node-level homophily metric used in Figure 1a, rather than the graph level. The formula for computing node-level homophily is $\mathcal{H}_{node}^{v}= \frac{|\\{u \mid u \in \mathcal{N}\_{v}, y\_{u}=y\_{v}\\}\|}{d\_{v}}$.
>
> >**Weakness #4:** RHO achieves significant improvements on some datasets, but demonstrates competitive performance on Reddit and Amazon. What could be the possible reasons behind this?
>
>
> For the Amazon dataset, which exhibits strong one-class homophily (i.e., nearly all normal nodes tend to have consistently high homophily while anomalies have low homophily, see Figure 1a), competing methods like GGAD excel at modeling such evident cues of normal nodes, and so they can achieve very strong detection performance.  Owing to the robust homophily learning capability, our method RHO is more advantageous in GAD datasets with complex homophily relations in the normal nodes, such as Elliptic that shows diverse homophily strengths in the normal nodes (see Figure 1a).  As a result, RHO can significantly outperform GGAD on datasets like Elliptic, but it may not be able to outperform GGAD on the datasets like Amazon that have monotonous homophily in the normal nodes.  For the Reddit dataset, anomaly detection is particularly challenging due to the extremely high global average similarity among the nodes in this dataset, which is close to 0.99, making it very difficult to differentiate anomalies from normal nodes from various aspects. As a result, most baseline methods achieve an AUROC of only around 0.55; although GGAD and RHO perform better on this dataset, both still find it difficult to put anomalies at the top in the anomaly ranking. Modeling more discriminative anomaly/normality priors may be needed for more promising performance on Reddit.

---

> > ### Comment · Reviewer_tFPZ · 2025-08-04
> >
> > Thanks for the authors' responses, which have addressed my concerns and further improved the paper's quality. Therefore, I would like to increase my rating.

---

> > > ### Author Response · Authors · 2025-08-04
> > > **Response to Reviewer tFPZ**
> > >
> > > Dear Reviewer tFPZ,
> > >
> > > Thank you very much for your thoughtful comments and for raising your rating. We're delighted that our responses have addressed your concerns. Please kindly let us know if there are any further questions. Thanks again!
> > >
> > > Best regards,
> > >
> > > Authors

---

### Official Review · Reviewer_pqei · 2025-07-01

**Clarity:** 3
**Significance:** 2
**Originality:** 3
**Rating:** 4
**Confidence:** 3

**Summary:**

The paper proposes RHO, a semi‐supervised graph anomaly detection (GAD) method that learns heterogeneous homophily patterns from a small set of labeled normal nodes. RHO combines two novel modules:
1. AdaFreq: adaptive spectral filters with trainable frequency‐response parameter $k$, applied in both cross‐channel and channel‐wise views to capture diverse low‐ and high‐frequency components of normal nodes.
2. Graph Normality Alignment (GNA): a contrastive alignment loss that enforces consistency between the two views’ representations.

**Questions:**

See weaknesses above

**Ethical Concerns:**

["NO or VERY MINOR ethics concerns only"]

**Final Justification:**

The rebuttal has addressed most of my concerns, thus I am inclined to weak accept this paper.

**Limitations:**

The proposed method only applies to node-level anomaly detection. Also, the authors do not provide interpretation of the learned $k$.

**Quality:**

3

**Strengths And Weaknesses:**

### **Strengths**
The proposed method:
1. Leveraging both cross‐channel and channel‐wise perspectives captures complementary homophily patterns.
2. The GNA module effectively aligns the two views, improving robustness.

### **Weaknesses**

#### **1. The analyses are confusing at times:**

**In Theorem 1:**

1.  **"assuming $β_m > 0$":** It is unclear why all projection coefficients \(\beta_m\) must be positive. What if \(\beta_m<0\)? Does the result still hold or collapse? Any justification or empirical evidence for this assumption is missing.
2.  **Motivation for the filter design:** The manuscript states “According to Theorem 1, … by training the parameter $k$.” However, Theorem 1’s statement involves $g(\lambda_m)$ under an assumption on $\beta_m$ The definition of $g(\lambda)$in Theorem 1 differs from Eq. (2) (where $g(\lambda)=1-k\lambda$), so it is unclear how Theorem 1 *motivates* the choice of this linear filter form.
3. **"When k = 0, g(λ) retains raw frequency signals.":** The claim “When $k=0$, $g(\lambda)$ retains raw frequency signals” seems contradictory: if $g(\lambda)=1$, then the output is constant across all eigenvalues, not the original signal $\lambda$. Should raw frequency signal be $\lambda$ rather than 1?

#### **2. Adaptive filters are well-studied in the graph learning field. What's new (advantages) in your method over existing ones?**

### Minor Issues
1. “which is simultaneously applies on” → should be “which is simultaneously applied to” (Sec. 4.1).
2. “AUROCassesses the model’s overall ability...” → missing space: “AUROC assesses…” (Sec. 5.1).

---

> ### Author Rebuttal · Authors · 2025-07-31
>
> We sincerely appreciate your constructive and positive comments on our model design. Please see our response to your comments one by one below.
>
> >**Weakness #1.1:** "assuming $\beta_m>0$": It is unclear why all projection coefficients ($\beta_m$) must be positive. What if ($\beta_m$<0)? Does the result still hold or collapse? Any justification or empirical evidence for this assumption is missing.
>
>
> Thank you for raising the question regarding the assumption of $\beta_m>0$. For simplicity, we only analyze one side of the symmetric case in the derivation; when the $m$-th frequency components exhibit high consistency across the labeled normal nodes, e.g., $u_m(i)$ have the same sign (all positive or all negative) and similar amplitude for $i\in \mathcal{V}\_l$, then the ratio $\frac{\sum_{i \in \mathcal{V}\_l }u_m(i)}{\beta_m\sum_{i \in \mathcal{V}\_l }u_m(i)^2}$ will be either much greater than 0 (case 1: $\beta_m>0$ and $\frac{\sum_{i \in \mathcal{V}\_l }u_m(i)}{\sum_{i \in \mathcal{V}\_l }u_m(i)^2} \gg 0$, case 2: $\beta_m<0$ and $\frac{\sum_{i \in \mathcal{V}\_l }u_m(i)}{\sum_{i \in \mathcal{V}\_l }u_m(i)^2} \ll 0$)
> or much less than 0 (case 1: $\beta_m>0$ and $\frac{\sum_{i \in \mathcal{V}\_l }u_m(i)}{\sum_{i \in \mathcal{V}\_l }u_m(i)^2} \ll 0$, case 2: $\beta_m<0$ and $\frac{\sum_{i \in \mathcal{V}\_l }u_m(i)}{\sum_{i \in \mathcal{V}\_l }u_m(i)^2} \gg 0$), resulting in a frequency positively or negatively enhancing response $g(\lambda_m) \gg 0$ or $g(\lambda_m) \ll 0$.
> Only when the frequency components are highly inconsistent, e.g., $u_m(i)$ exhibits opposite signs across the labeled normal nodes, these inconsistent components will cancel each other out (i.e., $\sum_{i \in \mathcal{V}\_l }u_m(i) \rightarrow 0$), resulting in a frequency suppressing response $g(\lambda_m) \to 0$. To summarize, our theoretical conclusion—minimizing the one-class loss enables an adaptive filtering mechanism to automatically enhance frequency components that are consistent across normal nodes while suppressing the inconsistent components—does not depend on the sign of $\beta_m$. Instead, it is solely determined by the consistency of the frequency components on the labeled normal nodes. We will clarify this in the final version.
>
> >**Weakness #1.2:** Motivation for the filter design: The definition of $g(\lambda)$ in Theorem 1 differs from Eq. (2) (where $g(\lambda) = 1-k\lambda$), so it is unclear how Theorem 1 motivates the choice of this linear filter form.
>
> Thank you very much for the comment. $g(\lambda)$ is a graph spectral filter function defined with respect to the eigenvalue $\lambda$. In Theorem 1, we unify the graph spectral filter with the one-class learning objective function to theoretically characterize the spectral behavior of the filter $g(\lambda)$ during optimization. So, $g(\lambda)$ in Theorem 1 is not a definition of the filter function itself, but rather a theoretical description of its desired frequency response. Theorem 1 motivates us to design an adaptive filter that dynamically adjusts the response strength of different frequency, effectively preserving the consistent frequency components of labeled normal nodes with diverse homophily in the optimization. In our implementation, we adopt a linear filter formulation due to its simplicity and effectiveness. Specifically, we parameterize the filter as $g(\lambda) = 1-k\lambda$ where the parameter $k$ is learned to ensure that the filter adheres to the underlying homophily pattern. Moreover, when stacking $K$ layers, the proposed filter becomes $g(\lambda) = \prod_{i=1}^{K} (1 - k_i \lambda)$, which can be further expressed as a polynomial expansion, i.e., $g(\lambda) = \sum_{l=0}^{K} \alpha_l \lambda^l$, where $\alpha_l$ are learnable parameters derived from $k_i$. Therefore, our filter also possesses nonlinear expressive capability when the network layer $K>1$.
>
> We further compared our filter with an adaptive nonlinear spectral filter $h(\lambda) = a_0 \lambda + \sum_{i=1}^{d/2} a_i sin(\frac{\epsilon \lambda}{10000^{2i/d}}) + \sum_{i=1}^{d/2} b_i  cos(\frac{\epsilon \lambda}{10000^{2i/d}})$ based on Fourier series, as used in the SpecFormer method [Ref1]. In the experiments, we replaced the filter function $g(\lambda)$ in the RHO framework with SpecFormer’s nonlinear filter $h(\lambda)$ and conducted the experiments under the same settings. The results are presented in Tables A1 and A2.
>
> ```
> Table A1. AUROC results of our filter and SpecFormer’s nonlinear filter.
> ```
> |Datasets|Reddit|Tolokers|Amazon|
> |---|---|---|---|
> |SpecFormer|0.5923|0.6017|0.9039|
> |RHO|**0.6207**|**0.6255**|**0.9302**|
>
> ```
> Table A2. AUPRC results of our filter and SpecFormer’s nonlinear filter.
> ```
> |Datasets|Reddit|Tolokers|Amazon|
> |---|---|---|---|
> |SpecFormer|0.0493|0.3073|0.7110|
> |RHO|**0.0616**|**0.3256**|**0.7879**|
>
> From the results, our proposed filter demonstrates superior performance compared to the complex nonlinear spectral filter used in SpecFormer. The results indicate that our simple linear design exhibits stronger adaptability and better generalization in capturing homophilic patterns among normal nodes. In contrast, the complex filter design in SpecFormer is likely to overfit the limited number of normal samples in the training set,  resulting in less effective generalization. These findings further validate the effectiveness and generality of our filter in graph anomaly detection. In addition, the nonlinear filter in SpecFormer is a type of filter that typically requires explicit spectral decomposition, which incurs a computational cost of $O(N^3)$, making it not scalable to large-scale graphs.
>
> - [Ref1] Bo, D., et al.  Specformer: Spectral graph neural networks meet transformers. ICLR 2023.
>
> >**Weakness #1.3:** "When $k$ = 0, $g(\lambda)$ retains raw frequency signals.": The claim ''when $k=0, g(\lambda)$ retains raw frequency signals'' seems contradictory: if $g(\lambda)=1$, then the output is conatant across all eigenvalues, not the original signal $\lambda$. Should raw frequency signal be $\lambda$ rather than 1?
>
> We apologize for the misunderstanding caused by this concept. We would like to clarify that the "signal" in this sentence refers specifically to the graph signal $\mathbf{x} \in \mathbb{R}^{N}$. As stated in the first sentence of our theoretical analysis, ''A graph filter operation on a graph signal $\mathbf{x} \in \mathbb{R}^{N}$ can be defined as $\mathbf{z} = \mathbf{U}g(\mathbf{\Lambda})\mathbf{U}^\top\mathbf{x}$.''  When $k$ = 0, the filter $g(\lambda) = 1$, which is equivalent to $g(\mathbf{\Lambda}) = \mathbf{I}$. As a result, we obtain $\mathbf{z} = \mathbf{U}\mathbf{I}\mathbf{U}^\top\mathbf{x}=\mathbf{x}$, indicating that the filter $g(\lambda)$ leaves the signal unchanged and thus preserves the raw input $\mathbf{x}$. We will revise the content and provide this detailed clarification in the final version.
>
> >**weakness #2:** Adaptive filters are well-studied in the graph learning field. What's new (advantages) in your method over existing ones?
>
> Adaptive filters are well-studied for addressing the heterophily issue in node classification tasks. But for graph anomaly detection, i) its one-class classification nature poses significant challenges for the application of conventional spectral methods; and ii) moreover, we empirically reveal that the normal nodes can have diverse homophily in the GAD datasets, which has been overlooked by existing methods. The adaptive filter we propose has a unique advantage that it adaptively captures a wide range of homophily patterns among the nodes in the normal class. Such a filter is specifically designed to address the two aforementioned issues in graph anomaly detection, offering important insights into the one-class homophily learning problem.
>
> >**Minor Issues**
> “which is simultaneously applies on” → should be “which is simultaneously applied to” (Sec. 4.1).
> “AUROC assesses the model’s overall ability...” → missing space: “AUROC assesses…” (Sec. 5.1).
>
> Thanks for pointing out these typo errors. We will correct them in the final version.
>
> >**Limitation:** The proposed method only applies to node-level anomaly detection. Also, the authors do not provide an interpretation of the learned $k$.
>
> Thank you very much for the comments. Node-level anomaly detection is among the most popular and challenging GAD tasks due to its broad real-world applications in diverse domains. Other GAD tasks, such as edge-level or graph-level anomaly detection, require very different designs, since we are required to deal with different types of graph instances, such as a set of edges or full graphs. Thus, similar to numerous existing studies, we focus on node-level anomaly detection in this work and plan to explore the proposed methodology in edge- or graph-level anomaly detection in our future work.
>
> To enhance the interpretation of the learnable $k$ in our adaptive filter, we have provided an intuitive visualization in Figure 4 in Section 5.2. The results show that in real-world datasets, the cross-channel view learns positive $k$ values, which approximate the low-pass filtering curve (inversely proportional to the frequency $\lambda$), supporting the modeling of low-frequency signals. Meanwhile, in the channel-wise view, negative $k$ values  are learned for different feature channels, which approximate high-pass filter curves (proportional to the frequency $\lambda$) capturing the high-frequency signals; in some cases we obtain $k=0$, which approximates horizontal all-pass filter curve, preserving the raw signals. We will clarify this interpretation of $k$ in the final version.

---

> > ### Comment · Reviewer_pqei · 2025-08-05
> > **Thanks for the rebuttal**
> >
> > I've reviewed your response, and it addresses most of my concerns. Please ensure that the points we discussed are fully incorporated into your revised version.
> >
> > Best regards

---

> > > ### Author Response · Authors · 2025-08-05
> > > **Response to Reviewer pqei**
> > >
> > > Dear Reviewer pqei,
> > >
> > > Thank you very much for your positive feedback on our response. We're pleased to know that our replies have addressed most of your concerns. We will carefully prepare the final version of our paper and fully incorporate the additional results and clarifications into the revised paper. Please kindly let us know if there are any further questions. Thanks again!
> > >
> > > Best regards,
> > >
> > > Authors

---

### Official Review · Reviewer_Yk4n · 2025-07-02

**Clarity:** 3
**Significance:** 3
**Originality:** 3
**Rating:** 4
**Confidence:** 4

**Summary:**

This paper proposes a novel semi-supervised graph anomaly detection framework named RHO, which aims to address the challenge of homophily among normal nodes—i.e., the fact that normal nodes may have either high or low homophily in graph structure. To tackle this, the authors introduce AdaFreq, an adaptive frequency-response filter that adjusts graph signal propagation strengths based on learnable frequency parameters, applied in both a cross-channel view (aggregating all features) and a channel-wise view (learning distinct filters per feature channel). To unify these two complementary views, the paper further proposes a Graph Normality Alignment (GNA) module that encourages consistent representations across views through a contrastive alignment objective. Experimental results and visualizations demonstrate that RHO effectively captures diverse normal patterns and improves anomaly detection robustness compared to existing methods.

**Questions:**

1. In the channel-wise view, each feature dimension is propagated independently using a dedicated frequency filter. Have the authors considered modeling interactions between feature dimensions (e.g., pairwise or grouped channels)? Would such multi-dimensional interactions further enhance the model’s capacity to capture complex homophily patterns?

2. In the experiments, most baseline methods are from earlier years, with only a few from 2024.

3. The GNA module focuses on aligning representations across the cross-channel and channel-wise views. Within the channel-wise view, however, each feature dimension is processed independently. Is there any mechanism to encourage consistency or coordination among individual feature channels?

**Ethical Concerns:**

["NO or VERY MINOR ethics concerns only"]

**Final Justification:**

We thank the authors for their detailed rebuttal, which clearly addressed the concerns we raised—especially Question 1 regarding channel interactions.

We encourage the authors to ensure that all discussed revisions and clarifications are carefully integrated into the final version.

Good luck!

**Limitations:**

See above.

**Paper Formatting Concerns:**

No formatting concerns.

**Quality:**

3

**Strengths And Weaknesses:**

**Strengths:**

1. **Novel problem formulation**: The paper identifies an important yet underexplored issue in graph anomaly detection named *heterogeneous homophily among normal nodes*, and it designs the model explicitly to address this challenge.

2. **Methodological originality**: The introduction of AdaFreq, a learnable frequency-response filter applied to both cross-channel and channel-wise views, is a compelling design. Especially, using channel-wise frequency modulation to model per-feature homophily variation is a fresh perspective.

3. **Effective view alignment strategy**: The proposed Graph Normality Alignment (GNA) module employs a contrastive objective to reconcile representations from different views, leading to more robust normality modeling.

4. **Strong empirical results**: The paper presents comprehensive experiments with both quantitative metrics and qualitative visualizations (e.g., t-SNE, anomaly score histograms), which together provide strong support for the method’s effectiveness.

5. **Clarity of exposition**: The paper is clearly written and logically structured. Design motivations are well-justified, and ablation studies are provided to isolate the impact of each module.

**Weaknesses:**

1. While the paper focuses on per-feature adaptive filtering, it does not explore potential interactions across multiple feature combinations (e.g., pairwise or group-wise channel dependencies), which may limit its capacity to capture complex homophily patterns.

2. Some modules, such as GNA, introduce additional complexity. A more detailed runtime or scalability analysis would further strengthen the paper.

---

> ### Author Rebuttal · Authors · 2025-07-31
>
> We sincerely appreciate your constructive and positive comments on the presentation, visualization, method, and empirical justification. Please see our response to your comments one by one below.
>
> >**Weakness #1 and Question #1:** While the paper focuses on per-feature adaptive filtering, it does not explore potential interactions across multiple feature combinations (e.g., pairwise or group-wise channel dependencies), which may limit its capacity to capture complex homophily patterns.
>
> Thanks for your valuable suggestion. The cross-channel filter was designed in RHO to capture the interactions among part/all of the channels. To examine the necessity of including additional interactions beyond our cross-channel design,  we add group-wise channels by enumerating all possible groups of a fixed size to explore more potential interactions. We then learn a dedicated filter for each group. Note that  when the group size is set to two, the group-wise view essentially reduces to the pairwise view. To ensure that the group-wise channel dependencies align well with the RHO framework, we concatenate the representations obtained from both the pairwise/group-wise and cross-channel views and feed it to the one-class classification loss $\mathcal{L}_{ccr}$. The fused representation is then aligned with the channel-wise representation using the proposed GNA module. Tables A1 and A2 present the performance of two group configurations with different group sizes (gs) across five benchmark datasets.
>
>
> ```
> Table A1. AUROC results under varying group sizes on five datasets.
> ```
> ||Reddit|Tolokers|Amazon|Elliptic|Question|
> |---|---|---|---|---|---|
> |gs=2|0.5976|0.5199|0.9149|0.8469|0.5514|
> |gs=3|0.5813|0.5451|0.9055|0.8195|0.5338|
> |RHO|**0.6207**|**0.6255**|**0.9302**|**0.8509**|**0.5833**|
>
> ```
> Table A2. AUPRC results under varying group sizes on five datasets.
> ```
> ||Reddit|Tolokers|Amazon|Elliptic|Question|
> |---|---|---|---|---|---|
> |gs=2|0.0505|0.2690|0.7478|0.4615|0.0423|
> |gs=3|0.0473|0.2759|0.6038|0.3577|0.0321|
> |RHO|**0.0616**|**0.3256**|**0.7879**|**0.5095**|**0.0430**|
>
> The experimental results indicate that incorporating the group-wise learning does not bring additional benefits to the model’s performance. Due to the lack of an informative grouping strategy, it can be difficult for our group-wise views to provide additional important information that our cross-channel view does not have for the model. By contrast, these group-wise views may contribute noisy or redundant information into the model optimization, which leads to less effective performance than the original RHO.
>
> In addition, please note that the use of these additional group-wise views can add a significant computational burden since there are a large number of such views in each dataset.
>
> >**Weakness #2:** Some modules, such as GNA, introduce additional complexity. A more detailed runtime or scalability analysis would further strengthen the paper.
>
> Thank you very much for the valuable suggestion. As discussed in Appendix E regarding time complexity,  although GNA may bring some additional complexity,  this overhead is acceptable given the performance gains compared with methods like DOMINANT (due to reconstruction) and TAM or GGAD (due to affinity calculations) that typically incur a time complexity of  $O(N^2)$,  causing their runtime to increase significantly as the number of nodes grows. To demonstrate this, we further provide a running time comparison result in Table A3. Although our method can run on all datasets using an RTX 3090 GPU, some baseline methods may encounter out-of-memory (OOM) issues on the same GPU due to their different computational mechanisms, highlighting that RHO is more memory-efficient. To ensure a fair comparison, we report the runtime performance on the CPU for all methods. The runtime results demonstrate that RHO remains efficient even on the large-scale graph T-Finance, primarily due to the use of mini-batch processing in the GNA module. By adopting a mini-batch strategy, the complexity of GNA reduces to $O(b^2d)$ for a batch size of $b$, and the total per-epoch complexity becomes $O(Nbd)$ when averaged across all batches, which significantly reduces memory usage and computational overhead, thereby improving the scalability. In contrast, methods such as DOMINANT, TAM, and GGAD involve reconstruction or affinity calculations across all nodes, which typically incur significant computational overhead on large-scale graphs.
>
> ```
> Table A3. Runtimes (in seconds, including both training and testing runtimes) on the five datasets.
> ```
> ||Reddit|Photo|Amazon|Question|T-Finance|
> |---|---|---|---|---|---|
> |DOMINANT|125|437|1592|740|10721|
> |OCGNN|162|125|765|973|5717|
> |AEGIS|166|417|1121|660|15258|
> |TAM|432|165|4516|29280|17360|
> |GGAD|368|136|1020|1125|9345|
> |ConsisGAD|483|681|3259|483|20172|
> |RHO|398|201|3358|675|5184|
>
>
>
> >**Question #2:**  In the experiments, most baseline methods are from earlier years, with only a few from 2024.
>
> Thank you for your valuable suggestion. In response to **Reviewer yntr**’s comment, we have incorporated two additional baseline methods published in 2024. For detailed information regarding these newly added comparison, we kindly refer you to our response to **Reviewer yntr** – **Weakness #1**.
>
> >**Question #3:** The GNA module focuses on aligning representations across the cross-channel and channel-wise views. Within the channel-wise view, however, each feature dimension is processed independently. Is there any mechanism to encourage consistency or coordination among individual feature channels?
>
> Thank you for the valuable comments. We would like to clarify that the cross-channel is specially designed to explore the consistency and coordination among the feature channels by learning a shared filter for them.
>
> In addition, we have also explored the group-wise and pair-wise feature channels in our response to **Question #1**, and we empirically found that they do not bring additional benefits to RHO, indicating that the current cross-channel design can provide sufficient channel-interaction-based information that is complementary to the channel-wise view.

---

### Official Review · Reviewer_yntr · 2025-07-03

**Clarity:** 2
**Significance:** 3
**Originality:** 3
**Rating:** 5
**Confidence:** 4

**Summary:**

This paper introduces RHO, a robust homophily learning framework for semi-supervised graph anomaly detection that adaptively captures diverse normal patterns by learning spectral filters tailored to varying homophily levels. It proposes two key modules—AdaFreq, which learns channel-wise and cross-channel adaptive frequency-response filters, and GNA, which aligns their representations via contrastive learning to ensure consistency. Extensive experiments on eight real-world datasets show that RHO consistently outperforms state-of-the-art methods in detecting anomalies across heterogeneous homophily settings.

**Questions:**

See Weaknesses.

**Ethical Concerns:**

["NO or VERY MINOR ethics concerns only"]

**Final Justification:**

The rebuttal has addressed all my concerns.

**Limitations:**

See Weaknesses.

**Quality:**

3

**Strengths And Weaknesses:**

Strengths:
- The RHO framework’s adaptive spectral filters effectively capture diverse normal patterns across varying homophily levels.
- The integration of AdaFreq and GNA modules ensures both frequency-wise adaptation and representation consistency through contrastive learning.
- Evaluation on eight real-world datasets demonstrates RHO’s superior anomaly-detection performance.

Weaknesses:
- The paper is missing recent baselines such as PMP [1] and ConsisGAD [2].
- The term "cross-channel" is confusing: in this paper it refers to sharing across channels, but readers may interpret it as fusion among channels.
- Important sections, such as the complexity analysis, should be included in the main text rather than relegated to the appendix.

[1] Wei Zhuo, Zemin Liu, Bryan Hooi, Bingsheng He, Guang Tan, Rizal Fathony, and Jia Chen. "Partitioning Message Passing for Graph Fraud Detection." In The Twelfth International Conference on Learning Representations.

[2] Nan Chen, Zemin Liu, Bryan Hooi, Bingsheng He, Rizal Fathony, Jun Hu, and Jia Chen. "Consistency training with learnable data augmentation for graph anomaly detection with limited supervision." In The Twelfth International Conference on Learning Representations. 2024.

---

> ### Author Rebuttal · Authors · 2025-07-31
>
> We sincerely appreciate your constructive and positive comments on our design and empirical justification. Please see our response to your comments one by one below.
>
> >**Weakness #1:** The paper is missing recent baselines such as PMP [Ref1] and ConsisGAD [Ref2].
>
> Following your suggestion, we include two additional baselines, PMP and ConsisGAD, both published in 2024.  It's worth mentioning that both methods are fully supervised, assuming both the normal label and the abnormal label are known.  To ensure a fair comparison, similar to the adaptation of BWGNN, we adapt these methods to a semi-supervised setting by replacing the cross-entropy loss with the one-class classification loss. The experimental results are shown in Tables A1 and A2.
>
> ```
> Table A1. AUROC results of comparing RHO with two additional baselines on eight datasets.
> ```
> |Datasets|Reddit|Tolokers|Photo|Amazon|Elliptic|Question|T-Finance|DGraph|
> |---|---|---|---|---|---|---|---|---|
> |PMP|0.5472|0.5815|0.5844|0.8329|0.5617|0.5790|0.8321|0.5376|
> |ConsisGAD|0.5347|0.5974|0.5859|0.8715|0.7354|0.5737|0.8277|0.5735|
> |RHO|**0.6207**|**0.6255**|**0.7129**|**0.9302**|**0.8509**|**0.5833**|**0.8623**|**0.6033**|
>
> ```
> Table A2. AUPRC Results of comparing RHO with two additional baselines on eight datasets.
> ```
> |Datasets|Reddit|Tolokers|Photo|Amazon|Elliptic|Question|T-Finance|DGraph|
> |---|---|---|---|---|---|---|---|---|
> |PMP|0.0388|0.2939|0.1254|0.3582|0.1006|0.0383|0.4084|0.0046|
> |ConsisGAD|0.0360|0.3059|0.1319|0.6523|0.1765|0.0424|0.4152|0.0050|
> |RHO|**0.0616**|**0.3256**|**0.2337**|**0.7879**|**0.5095**|**0.0430**|**0.4893**|**0.0059**|
>
> From the results, we can observe that RHO consistently outperforms the two additional baselines across all benchmark datasets, reinforcing the better effectiveness and generalization of RHO against various competing methods. PMP and ConsisGAD are designed for fully supervised GAD settings where labeled anomalies can offer critical abnormality information during the optimization; they become less effective when applied in the semi-supervised setting where no labeled anomalies are unavailable and only a small portion of labeled normal nodes is available.
>
> - [Ref1]  Zhuo, W., et al. Partitioning message passing for graph fraud detection, In ICLR 2024.
>
> - [Ref2] Chen, N., et al. Consistency training with learnable data augmentation for graph anomaly detection with limited supervision. In ICLR 2024.
>
> >**Weakness #2:** The term "cross-channel" is confusing: in this paper it refers to sharing across channels, but readers may interpret it as fusion among channels.
>
> Thank you very much for the comment. We understand that the term "cross-channel" may have different interpretations in the literature. In our work, we use it to specifically denote the shared information across the channels, rather than channel fusion. ``Inter-channel'' may be a more precise term. We will use this alternative term and clearly define it within the context of our paper.
>
> >**Weakness #3:** Important sections, such as the complexity analysis, should be included in the main text rather than relegated to the appendix.
>
> Thank you for your valuable suggestion. We will move the complexity analysis to the main paper in the final version.

---

### Note · Authors · 2025-08-14

Dear Reviewers and AC,

We wish to express our sincerest gratitude to the AC and all reviewers for your hard work and insightful feedback throughout the review process.  In our rebuttal, we are dedicated to addressing all raised concerns by providing new experiments and analyses, including:

(1) **Clarification of theory and motivation**: We clarified our theoretical assumption and supplemented it with additional explanations and experimental results to further elaborate on the motivation behind our filter design, while also highlighting the advantages of our adaptive filter in the GAD task.

(2) **Exploration on additional feature interactions**: We explored pairwise or group-wise channel dependencies, and  observed that adding pairwise/group-wise views leads to less effective performance than the original RHO. It is mainly because such combinations may contribute noisy or redundant information to the model optimization, which leads to sub-optimal performance.

(3) **Expanded evaluation**: Following the reviewers’ suggestions, we added two additional state-of-the-art baselines in the comparison and conducted a comprehensive runtime analysis on datasets with varying scales.

Through our rebuttal,  Reviewers **tFPZ** and **pqei** indicated that our responses addressed their main concerns and did not have follow-up questions; Reviewers **yntr** and **Yk4n** did not post any further questions too. Therefore, we believe that all of the major concerns have been adequately addressed. We are fully committed to integrating all these new results, detailed analyses, and clarifications into the final version of our paper.  Thanks again to AC and the reviewers for your time and effort on our paper.

Best regards,

Authors of Paper #15822

---

### Decision · Program_Chairs · 2025-09-17

**Decision:**

Accept (poster)

**Comment:**

This paper proposes RHO (Robust Homophily Learning), a framework for semi-supervised graph anomaly detection that tackles the challenge of heterogeneous homophily among normal nodes. The method combines AdaFreq, adaptive frequency-response filters applied at both channel-wise and cross-channel levels, with Graph Normality Alignment (GNA), which enforces consistency across views via contrastive learning. Experiments on eight real-world datasets demonstrate consistent improvements over strong baselines, including newly added 2024 methods.

Strengths: The paper addresses an important and underexplored problem in GAD, introduces a novel and well-motivated methodological design, provides solid theoretical insights, and shows strong empirical performance with comprehensive ablations and runtime analysis. The writing is generally clear, and visualizations aid intuition.

Weaknesses: The initial version missed recent baselines and runtime details, though these were added in rebuttal. Some theoretical explanations and notations were confusing, and exploration of richer feature interactions was limited. The scope remains restricted to node-level detection.

Rebuttal/Discussion: Reviewers’ main concerns (theory clarity, missing baselines, feature interactions, runtime analysis) were carefully addressed with new experiments and clarifications. Two reviewers explicitly increased their ratings after rebuttal, and others confirmed that their concerns were resolved.

The paper makes a solid, original contribution to semi-supervised GAD with convincing results. While not strong enough for spotlight/oral due to scope and generality limitations, it is a clear accept.